# Overlapping communities detection through weighted graph community games

**Stefano Benati**[1], **Justo Puerto**[2], **Antonio M. Rodríguez-Chía**[3], **Francisco Temprano**[2]*

**1** Dipartimento di Sociologia e Ricerca Sociale, Università di Trento, Trento, Italy, **2** IMUS, Universidad de Sevilla, Sevilla, Spain, **3** Faculty of Sciences, Universidad de Cádiz, Puerto Real (Cádiz), Spain

* ftgarcia@us.es

**Data Availability Statement:** All relevant data are within the manuscript and its Supporting information files.

**Funding:** The authors of this research Stefano Benati, Antonio Manuel Rodríguez-Chía, Justo

## Abstract

We propose a new model to detect the overlapping communities of a network that is based on cooperative games and mathematical programming. More specifically, communities are defined as stable coalitions of a weighted graph community game and they are revealed as the optimal solution of a mixed-integer linear programming problem. Exact optimal solutions are obtained for small and medium sized instances and it is shown that they provide useful information about the network structure, improving on previous contributions. Next, a heuristic algorithm is developed to solve the largest instances and used to compare two variations of the objective function.

## Introduction

The community detection problem consists in partitioning the node set of a network, or a graph, in such a way that node subsets can be substantially interpreted as communities. The methods that are proposed in the literature so far differ on two main aspects: the first is how community is translated into mathematics terms, the second is how an algorithm is implemented to outcome communities. To make an example, the classic contribution of [1] defines as a community the group of nodes with an arc density greater than what expected by nodes random pairing, then it proposes a method to find communities based on spectral decomposition. It is beyond our possibility to mention all contributions and developments that followed that seminal paper, see [2] for a comprehensive survey, but we just focus on the two most important lines of research that motivate our contribution. The first innovation recognizes that in some cases it is too restrictive to impose a strict nodes partition, as some node may realistically belongs to more than one community. So, communities can overlap and the solution structure is a node *assignment* to communities rather than a strict *partition*. A seminal contribution about overlapping communities can be found in [3] and a summary about first findings can be found in [4]. The second innovation is to formulate community detection as optimization problems, with a clearly stated objective function and well defined constraints. For example, in [5], the modularity model is developed into quadratic integer programming, corresponding to the well-known maximum clique partitioning. Other contributions can be found in [6–8].

Puerto and Francisco Temprano acknowledge financial support by the Spanish Ministerio de Ciencia y Tecnología, Agencia Estatal de Investigación and Fondos Europeos de Desarrollo Regional (FEDER) with grant number: PID2020-114594GB-C21, and Junta de Andalucía with grant number: P18-FR-1422. The authors Stefano Benati, Antonio Manuel Rodríguez-Chía and Justo Puerto also acknowledge partial support from: NetmeetData: Ayudas Fundación BBVA a equipos de investigación científica 2019 with reference "COMPLEX NETWORK". The author Antonio Manuel Rodríguez-Chía also acknowledges the European Regional Development Fund via projects with grant numbers: FEDER-UCA18-106895 and TED2021-130875B-I00; and the Spanish Ministerio de Ciencia y Tecnología, Agencia Estatal de Investigación with grant number: PID2020-114594GB-C22. The funders had no role in study design, data collection and analysis, decision to publish, or preparation of the manuscript.

**Competing interests:** The authors have declared that no competing interests exist.

The objective function is merely a simple statistic that evaluates partitions or node assignments. As such, it can be used to compare alternative community structures and to decide what is the most meaningful. One of the most popular statistic is modularity, see [1]. Modularity is an index that, for a given partition, compares the arc density of a subset with the one that is obtained on the assumption of node random pairings. The highest the modularity, the most connected are the nodes within a community, allowing a clear substantial definition of what is a community. The extension of the modularity to the case of overlapping communities has been proposed in [9], using fuzzy membership functions that are optimized using the fuzzy-*c*-means algorithm. This method has been elaborated further in [10–13], where the standard modularity function is modified by node or arc weights, representing node affinity, fuzzy memberships, or other. An alternative version of the objective function proposed in [9] is presented in [14], fixing some biases of the original one. In [7], it is proposed to maximize the modularity function, but with some additional constraints that allow some nodes to belong to more than one community. These nodes are referred to as *bridges*.

In [15], communities are defined as stable coalitions of a cooperative game. In a cooperative game, a coalition is stable if every member does not take any advantage in leaving the coalition to obtain a better payoff elsewhere, so a community is based on the concept of a common interest. There is a large room to define this common interest through any game characteristic function, such as market, voting, matching games, and so on. To just consider the topological network properties, such as the arc density and the node common neighbors, in [15] a weighted graph community game is proposed, with arc weights defined on some peculiar topological indicators. Next, an objective function is proposed to discern between alternative community structures and a constructive heuristic is implemented to find them.

In our contribution, we formulate the problem of finding communities as stable coalitions proposed in [15], as a mixed-integer linear programming problem. In this way, taking advantage of existing software, we can calculate the optimal communities of that model without resorting to any heuristic consideration. As a result, we can evaluate the optimal solutions of that model without the biases due to the use of the heuristic. Indeed, we found that the communities proposed in [15] are far from the optimal ones and, unfortunately, optimal ones are inconsistent too, in the sense that they do not correspond to what empirically one expects to find out. As it will be discussed, we argue that the reason of the inconsistency is on how costs of the weighted graph community game are defined and therefore we proposed a correction to them. Our correction follows the spirit of the modularity function, [1], in which an actual value of a statistic is compared to an expected value in absence of any community structure. We will show that our correction is reliable and effective as, after many computational tests, we showed that our method can recognize the hidden community structure of the networks. As a by-product of our contribution, we note that our cost definition relies on the calculation of the expected value of some network statistics on the assumption that no community is embedded in the network. To have an accurate cost estimate, we elaborated a new theorem to calculate the exact value of these statistics and it is worth to note that this theorem may have an autonomous interest for other applications in which some exact probabilities can be applied, as the same seminal paper [1].

To summarize, the contributions of our paper are the following:

1. We provide a mathematical formulation of the method proposed by [15] to detect the overlapping communities of a network.

2. We show that the communities obtained with this methodology are not the real communities embedded in the network, but we proposed an amendment to the game cost function that correct the bias.

3. We propose a heuristic algorithm that can calculate the optimal communities when the exact method fails because of the network size.

4. We apply our new mathematical model to real and artificial test problems and we show its effectiveness and reliability.

The paper is organized in 4 sections. In the Introduction, we motivate the paper purpose and summarize its contribution. In Material and methods Section, we formally introduce the overlapping community detection problem and the methods proposed by [15]. There, we design the exact optimization model and observe the finding of inconsistent communities. In Subsection called Detecting overlapping communities as stable coalitions of a cooperative game, we propose an alternative definition of the costs of the weighted graph community game that leads to a different objective function of the optimization model. In Local Stability Exploration Subsection, we present a heuristic algorithm for solving our model for the cases in which the network size is too large to compute the exact solution in a reasonable amount of time. In Results and discussion Section, we compare the exact and heuristic algorithm and then we report some computational results of a controlled experiment on graphs generated according the method proposed in [16] and we show that our method recovers correctly the community structure. The paper ends with some concluding remarks and outlines for future research in the final section, namely Conclusion.

## Material and methods

### Detecting overlapping communities as stable coalitions of a cooperative game

In [15], a cooperative game on a weighted graph is defined to characterize overlapping communities. The nodes of a graph are considered as the players of a network game, and then the Shapley value is used to characterize stable coalitions, e.g. subsets of nodes in which no player has any incentive to leave. Specifically, the cooperative game $(V, \varphi)$ is defined on the weighted graph $G = (V, E)$, with $V = \{1, \ldots, n\}$, e.g. players are nodes labeled from 1 to $n$, weights $W_{ij}(\geq 0)$ are defined for any edge $(i, j) \in E$, then the game characteristic function is:

$$\varphi(S) = \sum_{\substack{i, j \in S \\ i < j}} W_{ij}, \ \text{for} \ S \subseteq V \ . \tag{1}$$

That is, the value of coalition $S$ is the weights sum of the edges of the subgraph induced by $S$. The model has been called Weighted Graph Community (WGC) Game in the aforementioned paper.

When a coalition $S \subseteq V$ is going to form, then the members $i \in S$ can calculate the gain that they can get from it, e.g. what is their share of the payoff $\varphi(S)$ that they can receive. A standard result of cooperative games is that the share that they can get is the Shapley value of the game restricted to $S$: For player $i$ and coalition $S$, $i \in S$, the Shapley value is:

$$\varphi_i(S) = \frac{1}{2} \sum_{\substack{j \in S \\ j \neq i}} W_{ij}.$$

Hence, the profit of player $i$ from coalition $S$ depends on the total weight of its connection with the other members of $S$.

In [15], a coalition is defined *stable* if no member of $S$ takes advantage from swinging from coalition $S$ to coalition $V \setminus S$. In mathematical terms it occurs if and only if:

$$\varphi_i(S) \geq \varphi_i((V \setminus S) \cup \{i\}), \quad \forall i \in S. \tag{2}$$

Actually, there are different definition of stable coalitions that can be found in the literature: *Stable coalition structures* are defined in [17, 18], while in [19, 20], condition (2) is called the *internal stability property*. Moreover, in the latter notion of stability, an additional property is imposed requiring that a coalition $S$ is stable if no member of $S$ takes advantage from swinging from $S$ to any other subset $S'$ contained in $V \setminus S$. This can be formalized as:

$$\varphi_i(S) \geq \varphi_i(S' \cup \{i\}), \quad \forall i \in S, \quad \forall S' \subseteq V \setminus S. \tag{3}$$

However, we are not developing this issue further and we will remain with definition (2).

Formulating a WGC game allows a formal definition of what are the feasible overlapping communities of a network: As a node can belong to more than one stable coalition, communities can overlap. However, a crucial feature of the model is the way in which weights $W_{ij}$ are defined. In [15], the following formula is proposed: Let $k_i$ be the *adjacency degree* of node $i$ (e.g. the number of nodes to which $i$ is connected through an arc), let $P_{ij} = \frac{1}{k_i} + \frac{1}{k_j}$ be defined as the *partition ratio* and let $CN_{ij} = (|\text{common neighbors of i and j}| + 1)P_{ij}$ be defined as the *neighbourhood ratio* of $i, j \in V$, then the weight of the arc $(i, j)$, $i \neq j$ is

$$W_{ij} = \begin{cases} \frac{CN_{ij} - P_{ij}}{4}, & \text{if} \quad k_i \geq 1, \ k_j \geq 1 \text{ and } (i,j) \notin E, \\ P_{ij}, & \text{if} \quad k_i = 1 \text{ or } k_j = 1 \text{ and, } (i,j) \in E, \\ 2CN_{ij} + P_{ij}, & \text{if} \quad k_i > 1 \text{ and } k_j > 1, \text{ and } (i,j) \in E, \\ 0, & \text{otherwise.} \end{cases} \tag{4}$$

The formula was proposed in [15] to consider the node similarity as dependent on both the direct and indirect links between $i$ and $j$. It is straightforward to observe that $W_{ij} \geq 0$, but this property has important consequences on the structure of the stable coalitions, as it will be discussed later. For the moment, we focus in the methodology to find all the stable coalitions of a networks. While in [15] a constructive method is proposed, that is, an heuristic technique with some ad-hoc adjustment to find stable coalitions, here we propose a mathematical programming approach in which all considerations about stability discussed in [15] are translated into an objective function and mathematical constraints. We will show that stable coalitions can be represented by linear constraints involving binary variables and then, using an appropriate objective function, stable coalitions can be determined by linear programming.

Let $n_c$ be the maximum number of communities to which a node can belong to (this is not a binding constraint to the model, since $n_c$ can be large enough to include all the feasible stable communities). For $i = 1, \ldots, n$ and $k = 1, \ldots, n_c$, the model variables are:

$$x_{ik} = \begin{cases} 1, & \text{if node } i \text{ belongs to community/coalition } S_k, \\ \\ 0, & \text{otherwise.} \end{cases}$$

For any $i, j = 1, \ldots, n$ such that $i < j$ and $k = 1, \ldots, n_c$:

$$
z_{ijk} = \begin{cases} 1, & \text{if nodes } i \text{ and } j \text{ both belongs to community/coalition } S_k, \\ \\ 0, & \text{otherwise.} \end{cases}
$$

The relationship between $x$- and $z$-variables is given by the logical/quadratic constraints $z_{ijk} = x_{ik}x_{jk}$ for all $i, j \in V$, $i < j$ and all $k = 1, \ldots, n_c$. Then, the quadratic constraint can be replaced by the linear constraints:

$$
z_{ijk} \le x_{ik}, \quad \forall i,j = 1, \ldots, n, \ i < j, \ k = 1, \ldots, n_c, \tag{5}
$$

$$
z_{ijk} \le x_{jk}, \quad \forall i,j = 1, \ldots, n, \ i < j, \ k = 1, \ldots, n_c, \tag{6}
$$

$$
x_{ik} + x_{jk} - z_{ijk} \le 1, \quad \forall i,j = 1, \ldots, n, \ i < j. \ k = 1, \ldots, n_c. \tag{7}
$$

Next, using binary $x$-variables, the stability condition (2) can be characterized by linear constraints too. First, for fixed $i$ and $k$, consider the quadratic inequality:

$$
x_{ik} \left( \sum_{\substack{j=1 \\ j \ne i}}^{n} x_{jk} W_{ij} - \sum_{\substack{j=1 \\ j \ne i}}^{n} \left(1 - x_{jk}\right) W_{ij} \right) \ge 0.
$$

If $x_{ik} = 1$, then $i$ belongs to coalition $S_k$, so that $S_k$ must be stable. For the stability, $i$-player's Shapley value from coalition $S_k$ must be greater than its Shapley value from the opposite coalition $(V \setminus S_k) \cup \{i\}$. The term $\sum_{\substack{j=1 \\ j \ne i}} x_{jk} W_{ij}$ is the Shapley value of coalition $S_k$, as all $j$'s such that $x_{jk} = 1$ are all the other players of coalition $S_k$. Conversely, all other $j$'s such that $(1 - x_{jk}) = 1$ are the players excluded from $S_k$. Consequently, $\sum_{j=1}^{n} (1 - x_{jk}) W_{ij}$ is the Shapley value of the opposite coalition, $(V \setminus S_k) \cup \{i\}$. Finally, their difference must be greater than or equal to 0 for $S_k$ to be stable. Next, the above quadratic inequality can be simplified to the following linear one:

$$
\sum_{\substack{j=1 \\ j \ne i}}^{n} x_{jk} W_{ij} \ge \frac{\sum_{\substack{j=1 \\ j \ne i}}^{n} W_{ij} x_{ik}}{2}, \quad \forall i = 1, \ldots, n, k = 1, \ldots, n_c. \tag{8}
$$

Next, it must be imposed that overlapping coalitions/communities must have non-empty difference, e.g. the same coalition is not selected more than once (a coalition must not be contained in a different one). To prevent inclusion, additional variables $h$ are introduced for $i = 1, \ldots, n$ and pairs $k, r$ such that $1 \le k < r \le n_c$:

$$
h_{ikr} = \begin{cases} 1, & \text{if } i \text{ belongs to community } S_r \text{ and not to community } S_k, \\ \\ 0, & \text{otherwise.} \end{cases}
$$

The relation between $x$- and $h$-variables is given by the quadratic constraint: $h_{ikr} = x_{ir}(1 - x_{ik})$, that can be replaced by three linear constraints as done for $z$-variables in expressions (5)–(7).

To prevent the inclusion of $S_r$ in $S_k$, it must be that:

$$\sum_{j=1}^{n} h_{jkr} \geq x_{ir}, \quad \forall 1 \leq k < r \leq n_c, \forall i = 1, \dots, n. \tag{9}$$

The constraint is binding when $x_{ir} = 1$. In that case, coalition $S_r$ must contain at least one element $j$ that is contained in $S_r$ but not in $S_k$, guaranteeing that $S_r \not\subset S_k$.

To conclude, we introduce inequalities to avoid symmetrical solutions too. Symmetric solutions decrease the efficiency of the Integer Linear Programming solver, as the same structural solution can be obtained by multiple assignments to variables $x$, $z$, $h$, simply giving different labels to coalitions. Note that constraints (9) avoid to replicate the same coalition, so that it is sufficient that, after ranking the communities from the largest to the smallest, they are assigned to decreasing labels $k$. The following constraints do the task:

$$\sum_{i=1}^{n} x_{ik} \geq \sum_{i=1}^{n} x_{i,k+1}, \quad \forall k = 1, \dots, n_c - 1. \tag{10}$$

Every stable coalition corresponds to a point of the polytope described by the equations and inequalities described so far. To determine what are the most meaningful overlapping communities, in the objective function it is used the nodes Shapley value. If a coalition $S_k$ is established, then player $i$'s Shapley value from coalition $S_k$ is: $\sum_{j=1}^{n} z_{ijk} W_{ij}$. Therefore, for a set of overlapping communities $S_k$, $k = 1, \dots, n_c$, the total Shapley value of a player $i$ is the sum of the values it gets from every coalition, that is:

$$\sum_{k=1}^{n_c} \sum_{j=1}^{n} z_{ijk} W_{ij}. \tag{11}$$

In [15], the most important overlapping coalitions are determined by maximizing the sum of the Shapley values of all nodes. Therefore, this index will be used as the objective function of the following integer programming formulation:

$$(F_{Sh-JK}) \max \quad \sum_{k=1}^{n_c} \sum_{i=1}^{n-1} \sum_{j=i+1}^{n} z_{ijk} W_{ij} \tag{12}$$

s.t.: (5)–(10),

$$\sum_{k=1}^{n_c} x_{ik} \geq 1, \quad \forall i = 1, \dots, n, \tag{13}$$

$$h_{ikr} \leq 1 - x_{ik}, \quad \forall i = 1, \dots, n, \, k, r = 1, \dots, n_c, \, k < r, \tag{14}$$

$$h_{ikr} \leq x_{ir}, \quad \forall i = 1, \dots, n, \, k, r = 1, \dots, n_c, \, k < r, \tag{15}$$

$$x_{ir} - x_{ik} - h_{ikr} \leq 0, \quad \forall\, i = 1, \ldots, n,\ k, r = 1, \ldots, n_c,\ k < r, \tag{16}$$

$$x_{ik} \in \{0, 1\}, \quad \forall\, i = 1, \ldots, n,\ k = 1, \ldots, n_c, \tag{17}$$

$$z_{ijk} \in [0, 1], \quad \forall\, i, j = 1, \ldots, n,\ i \neq j,\ k = 1, \ldots, n_c, \tag{18}$$

$$h_{ikr} \in [0, 1], \quad \forall\, i = 1, \ldots, n,\ k, r = 1, \ldots, n_c,\ k < r. \tag{19}$$

The objective function (12) represents the sum of the Shapley values for all nodes and communities. Constraints (13) guarantee that every node belongs to at least one community. Constraints (14)–(16) are the linear representations of the $h$-variables. Finally, constraints (17) define binary variables. Note that in (18) and (19), we can relax the $z−$ and $h−$variables to be continuous, since the constraints on the $x$-variables force both to be binary.

$F_{Sh-JK}$ is the exact Integer Programming formulation of the model proposed in [15]. However, in that seminal paper the overlapping communities were computed through a heuristic constructive procedure, in which the search for optimal solutions is combined with various ad-hoc adjustments to induce sufficient diversification of coalitions. The advantage of Integer Programming is that the output coalitions of $F_{Sh-JK}$ are exactly the optimal ones, without any bias due to constructive rule-of-thumb procedures. As we will see, this allows us to point out a drawback of the game definition and to suggest a method to adjust it.

We apply formulation $F_{Sh-JK}$, to the Zachary's karate club network, fixing $n_c = 3$. Optimal overlapping communities can be seen in Fig 1. As can be seen, selected communities are the grand coalition (all the nodes belong to the same coalition) except one node. That is, communities are subsets $S$ such as $|S| = n - 1$, in which the discarded node is the one with less connections. It is hard to believe that those sets are of some interest to researchers, as they are far from the communities that were often identified in the Zachary's network. The same occurs

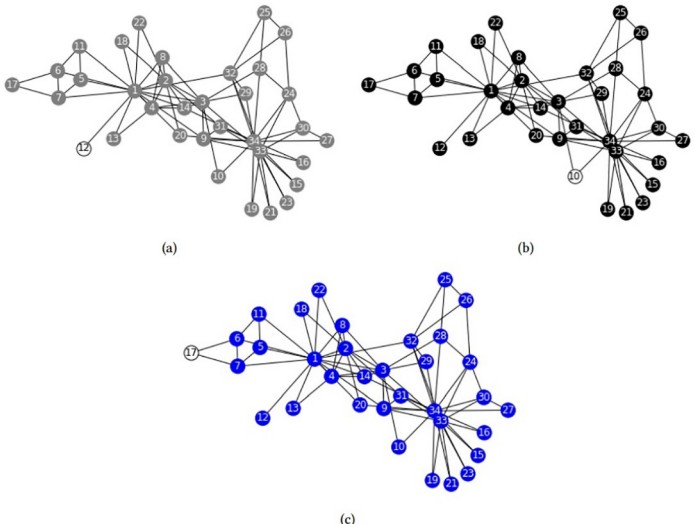

**Fig 1. Zachary's karate club structure obtained by $F_{Sh-Jk}$ with $n_c = 3$.** (a) Community 1, (b) Community 2, (c) Community 3.

with all the other problems we tested: Overlapping communities are the grand coalition except one node. The reason of this disappointing result is not the solution method, e.g. exact vs heuristic, or the community definition, e.g. using cooperative games and the Shapley value. Rather, the reason is the way in which weights $W$ are formulated in (4). As recognized in [15], if $W_{ij} \geq 0$ for all $i, j$, then the cooperative game $(V, \varphi)$ is convex, that is for two coalitions $S$, $T$ such that $S \subset T$ and $i \notin T$, it always occurs that:

$$\varphi(T \cup \{i\}) - \varphi(T) \geq \varphi(S \cup \{i\}) - \varphi(S).$$

This property establishes that the marginal gain player $i$ gets from joining a coalition is always greater when the coalition is larger. Therefore the Shapley values are always the greatest for the largest coalitions and that is why the method proposed is *always* doomed to mistake the largest subsets as communities. As we have pointed, the weakness is not on using cooperative games to define stable coalitions, but on using *convex* cooperative games. In this section, we will provide a simple and effective way to adjust this weakness. Our proposal is based on determining stability using a *non-convex* cooperative game.

**The computation of the expected weight on an arc.** As we discussed in the previous section, weighted graph community games in which arc weights $W_{ij} \geq 0$ are convex games, so that they imply increasing values of the Shapley values and the tendency of detecting only large size communities. A straightforward way of avoiding convexity is considering an alternative set of weights, non necessarily non-negative, so that optimal stable coalitions of small size may emerge as well. Here, we propose to combine the weights defined by (4) with modularity, so that weights are normalized by their expected values and may take both negative and positive values. As a consequence, the resulting game is non-convex.

The modularity function, see [1], is a well-known index to detect communities in networks. The index compares the edge density of the empirical graph $G = (V, E)$ (unweighted and undirected), $|E| = m$, with the expected edge density of a theoretical graph $G' = (V, E')$ in which there are no communities by assumption. The expected edge density of $G'$ is calculated using a null hypothesis, e.g. an assumption about the edge distribution, that is called the *configuration model*, [21]. If the graph does not contain communities, then for any given two nodes $i$ and $j$ with edge degrees $k_i$ and $k_j$, the expected number of edges between $i$ and $j$ is approximated by $\frac{k_i k_j}{2m}$. Let $A_{ij} = 1$ if $(i, j) \in E$, $A_{ij} = 0$ otherwise (so that $A = [A_{ij}]$ is the adjacency matrix of $G$). Moreover, let $\Pi$ be a partition of $V$ and let $\delta(i, j)$ be the Kronecker delta: $\delta(i, j) = 1$ if $i, j \in V$ belong to the same community, $\delta(i, j) = 0$ otherwise. Then the modularity function of a partition $\Pi$ is:

$$m(\Pi) = \frac{1}{2m} \sum_{i,j \in V} \left( A_{ij} - \frac{k_i k_j}{2m} \right) \delta(i, j). \tag{20}$$

In the case under study, weights are defined through expression (4), in which the adjacency between nodes $i$ and $j$ is weighted by the common neighbors. However, modularity can be defined for weighted graphs as well. In the summation terms $\left( A_{ij} - \frac{k_i k_j}{2m} \right)$, entries $A_{ij}$ are replaced by weights $W_{ij}$, $k_i$ replaced by weight sum $W_i = \Sigma_j W_{ij}$, and $m$ replaced by $W = \Sigma_{(i,j) \in E} W_{ij}$, as described in [22]. In this way, modularity is still a function that compares the actual indices of an empiric graph with the expected indices of a random graph. Using modularity, we can define modularity game $(V, \varphi)$ as a weighted graph community game in which the

characteristic function $\varphi$ is defined as in (1), but with the following weights:

$$W'_{ij} = W_{ij} - \frac{W_i W_j}{2W}. \tag{21}$$

In this case, $W'_{ij}$ can take both positive and negative values, so that the game resulting from the characteristic function (1) is non-convex.

We elaborate this model further, by noting that the modular term (21) should represent the difference between the empiric value $W_{ij}$ and its expected value under the assumption that the graph does not contain any communities. Unfortunately, the term $\frac{W_i W_j}{2W}$ is only an approximation of the true expectation and this can cause unexpected biases. For example, when weights $W_{ij}$ correspond to the adjacency matrix $A_{ij} \in \{0, 1\}$, the term $\frac{k_i k_j}{2m}$ is an estimate of the probability of an arc between $i$ and $j$, but, if the graph is unbalanced, the term can be greater than 1, which results in a non-sense estimation of this probability. In our application, expression (4) contains specific terms about the graph structure, such as the arcs and the common neighbours between two nodes, and potentially the bias between the true expectation and its approximation can be large. For this reason, we made a special effort in calculating the exact equation of the expected values of expression (4) under the assumption that there are no community in the graph.

In [21], the random occurrence of a graph with no communities is calculated through the configuration model. The configuration model can be interpreted as the process of making a random graph with no communities through the following operations. Every arc $e = (i, j)$ of the empirical graph $G = (V, E)$ is cut into two parts, say $l_1$ and $l_2$, with $l_1$ incident to $i$ and $l_2$ incident to $j$, called *stubs*. Next, two different stubs are selected randomly and paired. We say that, if $l_1$ and $l_2$ are such stubs, then $(l_1, l_2)$ is a match, e.g. an arc of the random graph $G' = (V, E')$. The way in which $G'$ is built implies that the adjacency degree $k_i$ remains unvaried for all $i$, but eventual communities are broken by random pairings of stubs. Note that, from construction, we can interpret any occurrence of $G'$ as a matching of $2m$ stubs. The process is exemplified in Fig 2.

Here we show how to compute exactly the expected values of expression (4) using the configuration model. Expected weights depend on the the partition ratio $P_{ij}$ and the neighbourhood ratio $CN_{ij}$ of the random graphs obtained from the configuration model. By construction, the partition ratio $P_{ij}$ of the random graph is the same as the one of the empiric graph, but the neighbourhood ratio $CN_{ij}$ is different.

To calculate $CN_{ij}$, we introduce some notation. Recall that $k_i$ is the adjacency degree of node $i$ and assume that the graph has $m$ edges. Let $P_{\text{adjacency}}(k_i, k_j, m)$ be the probability that node $i$ and $j$ are connected by an arc, let $P_{\text{common neighbour}}(k_i, k_j, k_r, m)$ be the probability that $i$ and $j$ are arc connected with $r$, so thar $r$ is a common neighbor, and let $P_{\text{triangle}}(k_i, k_j, k_r, m)$ be the probability that $i$ and $j$ are arc connected and are also connected with $r$, so that the three

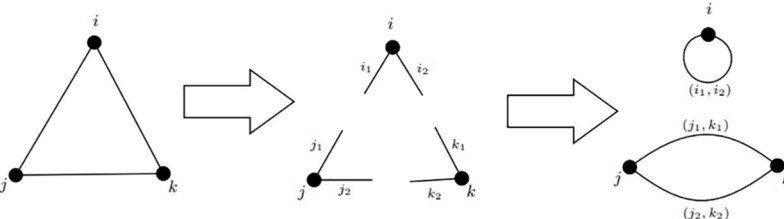

**Fig 2. Configuration model example.**

arcs form a triangle. The notation emphasizes that probabilities depend on adjacency degrees $k_i$, $k_j$, $k_r$ and the total number of edges $m$. In the following proposition, we will derive closed form expressions for the above probabilities.

**Proposition 1**. *Let i, j, r be three nodes with adjacency degrees $k_i$, $k_j$, $k_r$, respectively. Then, in the random graph configuration model*

$$P_{adjacency}(k_i, k_j, m) = \sum_{t=1}^{\min\{k_i, k_j\}} (-1)^{t+1} \frac{\binom{k_i}{t}\binom{k_j}{t}t!}{\prod_{p=1}^{t}(2m+1-2p)}, \quad (22)$$

$$P_{common\ neighbour}(k_i, k_j, k_r, m) = \sum_{t=1}^{\min\{k_j, k_r-1\}} (-1)^{t+1} P_{adjacency}(k_i, k_r - t, m - t) \frac{\binom{k_r}{t}\binom{k_j}{t}t!}{\prod_{p=1}^{t}(2m+1-2p)}, \quad (23)$$

$$P_{triangle}(k_i, k_j, k_r, m) = \sum_{t=1}^{\min\{k_i-1, k_j-1\}} (-1)^{t+1} P_{common\ neighbour}(k_i - t, k_j - t, k_r, m - t) \frac{\binom{k_i}{t}\binom{k_j}{t}t!}{\prod_{p=1}^{t}(2m+1-2p)}. \quad (24)$$

*Proof.* Applying the configuration model to $G = (V, E)$, we obtain two stubs $l_1$ and $l_2$, adjacent to $i$ and $j$, respectively, for every arc $e(i, j) \in E$. Then, we select two stubs at random and pair them until a random graph $G'$ is obtained. Note that, from construction, we can interpret any occurrence of $G'$ as a matching of $2m$ stubs.

Given $i, j \in V$, let $S_i = \{l_{i(1)}, \ldots, l_{i(ki)}\}$ be the set of stubs adjacent to $i$ and $S_j = \{l_{j(1)}, \ldots, l_{j(kj)}\}$ be the set of stubs adjacent to $j$. Assuming a set of $2m$ elements, there are $\frac{(2m)!}{2^m m!} = \prod_{p=1}^{m}(2m + 1 - 2p)$ different matching, see [23]. Therefore, if two stubs $l_1 \in S_i$ and $l_2 \in S_j$ are matched, there are $\prod_{p=1}^{m-1}(2m - 1 - 2p)$ different matching with the stubs remaining, because there are still $2m - 2$ stubs to pair. Due to this, the probability that two stubs $l_1$ and $l_2$ are joined, connecting nodes $i$ and $j$, is:

$$\frac{\prod_{p=1}^{m-1}(2m - 1 - 2p)}{\prod_{p=1}^{m}(2m + 1 - 2p)} = \frac{1}{2m - 1}. \quad (25)$$

Next, we introduce random variables:

$$X_{l_1 l_2} = \begin{cases} 1, & \text{if the stubs } l_1 \text{ and } l_2 \text{ are matched,} \\ \\ 0, & \text{otherwise.} \end{cases}$$

Obviously, the probability of $X_{l_1 l_2} = 1$ is $P(X_{l_1 l_2} = 1) = \frac{1}{2m-1}$, as stated in (25). We can express the number of edges between two nodes $i$ and $j$ as the sum:

$$\sum_{l_1 \in S_i} \sum_{l_2 \in S_j} X_{l_1 l_2}.$$

The above expression represents the sum of the variables $X_{l_1 l_2}$ whose indices are one stub adjacent to $i$ and another stub adjacent to $j$. Thus, the expected number of edges between $i$ and

$j$ is:

$$\sum_{l_1 \in S_i} \sum_{l_2 \in S_j} E[X_{l_1 l_2}] = \sum_{l_1 \in S_i} \sum_{l_2 \in S_j} P(X_{l_1 l_2} = 1) = \frac{k_i k_j}{2m - 1}.$$

Note that in the modularity function (20), this value is approximated by $\frac{k_i k_j}{2m}$.

As we explain before, the expected number of edges is different to the probability of adjacency. The adjacency between two nodes $i$ and $j$ is the condition that there is at least one arc between $i$ and $j$ and it can be expressed as the union of the events $\{\omega \ : \ X_{l_1 l_2}(\omega) = 1\}$ with $l_1 \in S_i$ and $l_2 \in S_j$, for the sake of simplicity, we refer to this set of events as $\{X_{l_1 l_2} = 1\}$. So, the adjacency probability of two nodes $i$ and $j$ is:

$$P\left( \bigcup_{\substack{l_1 \in S_i \\ l_2 \in S_j}} \left\{ X_{l_1 l_2} = 1 \right\} \right). \tag{26}$$

Let $S_{ij}^t$ be the set of all the different subsets of $S_i \times S_j$ with size $|S_{ij}^t| = t$. Applying the inclusion-exclusion law for the probability of union of events to expression (26), it follows that:

$$P\left( \bigcup_{\substack{l_1 \in S_i \\ l_2 \in S_j}} \{X_{l_1 l_2} = 1\} \right) = \sum_{t=1}^{k_i k_j} (-1)^{t+1} \sum_{S \in S_{ij}^t} P\left( \bigcap_{(l_1, l_2) \in S} \{X_{l_1 l_2} = 1\} \right). \tag{27}$$

By construction of the random graph $G'$, observe that the intersection of $t$ different sets $\{X_{l_1 l_2} = 1\}$, representing the match between stubs $l_1$ and $l_2$, is empty if the same stub, $l_1$ or $l_2$, is repeated more than once in different matches. Therefore, for each $t$, the non empty sets $\bigcap_{(l_1, l_2) \in S} \{X_{l_1 l_2} = 1\}$ that appears in (27) are matching with $t$ matches. As a consequence, the summation on $t$ is bounded to $\min\{k_i, k_j\}$, because the intersection of more than $\min\{k_i, k_j\}$ different sets must repeat some stubs and so, its intersection is empty. Moreover, applying the same argument to calculate the probability of joining two stubs (25), the probability of joining $t$ stubs from $S_i$ with other $t$ stubs from $S_j$ is:

$$\frac{\prod_{p=1}^{m-t}(2m + 1 - 2t - 2p)}{\prod_{p=1}^{m}(2m + 1 - 2p)} = \frac{1}{\prod_{p=1}^{t}(2m + 1 - 2p)}.$$

Finally, to derive expected vales, we need to calculate the number of different subsets from $S_i \times S_j$ with a size equal to $t$ that do not repeat any stubs. We have to consider $t$ stubs from $S_i$ and $t$ from $S_j$, and then all the possible matchings between stubs of different sets. There are $\binom{k_i}{t}$ different subsets of $t$ stubs from $S_i$ and $\binom{k_j}{t}$ different subsets of $t$ stubs from $S_j$. We can match the $t$ stubs of one set with the other $t$ stubs of the other set in $t!$ different ways, obtaining the following expression for the probability of events ensuring that node $i$ and $j$ are connected, in

short, $\{i \text{ and } j \text{ are connected}\}$:

$$P(\{i \text{ and } j \text{ are connected}\}) = P\left(\bigcup_{\substack{l_1 \in S_i \\ l_2 \in S_j}} \{X_{l_1 l_2} = 1\}\right) =$$

$$= \sum_{t=1}^{\min\{k_i,k_j\}} (-1)^{t+1} \frac{\binom{k_i}{t}\binom{k_j}{t}!}{\prod_{p=1}^{t}(2m+1-2p)}. \tag{28}$$

This is the expression in (22) for $P_{\text{adjacency}}(k_i, k_j, m)$.

Now, we use (28) and the previous arguments to obtain the probability that $i$ and $j$ are connected with a different node $r$, namely $P_{\text{common neighbour}}(k_i, k_j, k_r, m)$, i.e., we compute the probability of the intersection of the event nodes $i$ and $r$ are connected with the event nodes $j$ and $r$ are connected, in short, $\{i \text{ and } r \text{ connected}\} \cap \{j \text{ and } r \text{ connected}\}$:

$$P(\{i \text{ and } r \text{ connected}\} \cap \{j \text{ and } r \text{ connected}\}) = P\left(\{i \text{ and } r \text{ connected}\} \cap \bigcup_{\substack{l_1 \in S_i \\ l_2 \in S_j}} \{X_{l_1 l_2} = 1\}\right)$$

$$= \sum_{t=1}^{\min\{k_j,k_r-1\}} (-1)^{t+1} \sum_{S \in S_{jr}^t} P\left(\{i \text{ and } r \text{ connected}\} \cap \bigcap_{(l_1,l_2)\in S} \{X_{l_1 l_2} = 1\}\right)$$

$$= \sum_{t=1}^{\min\{k_j,k_r-1\}} (-1)^{t+1} \sum_{S \in S_{jr}^t} P\left(\{i \text{ and } r \text{ connected}\} \Big| \bigcap_{(l_1,l_2)\in S} \{X_{l_1 l_2} = 1\}\right) P\left(\bigcap_{(l_1,l_2)\in S} \{X_{l_1 l_2} = 1\}\right) \tag{29}$$

$$= \sum_{t=1}^{\min\{k_j,k_r-1\}} (-1)^{t+1} P_{\text{adjacency}}(k_i, k_r - t, m - t) \frac{\binom{k_r}{t}\binom{k_j}{t}t!}{\prod_{p=1}^{t}(2m+1-2p)}.$$

Finally, developing as before, the probability of three nodes $i$, $j$ and $r$ to be connected each other, namely $P_{triangle}(k_i, k_j, k_r, m)$ is:

$$P\left(\{i \text{ and } r \text{ connected}\} \cap \{j \text{ and } r \text{ connected}\} \cap \{i \text{ and } j \text{ connected}\}\right)$$

$$= \sum_{t=1}^{\min\{k_i-1,k_j-1\}} (-1)^{t+1} P_{\text{common neighbour}}(k_i - t, k_j - t, k_r, m - t) \frac{\binom{k_i}{t}\binom{k_j}{t}t!}{\prod_{p=1}^{t}(2m+1-2p)}. \tag{30}$$

The above probabilities are necessary to determine the exact value of the expected weight $E[W_{ij}]$, when weights are defined as in formula (4) and the graph is obtained by the configuration model.

Define the following random variables:

$$Y_{ij} = \begin{cases} 1, & \text{if nodes } i \text{ and } j \text{ are connected,} \\ 0, & \text{otherwise,} \end{cases} \quad \forall i, j \in V.$$

**Theorem 1**. *Assume that weights between nodes $i$ and $j$ are defined as in (4), then the expected weight $E[W_{ij}]$ between nodes $i$ and $j$ of the the random graph configuration model is given by the following expressions:*

1. If $k_i = 1$ or $k_j = 1$,

$$E[W_{ij}] = \frac{P_{ij}\sum_{r \in V \setminus \{i,j\}} P_{common\_neighbor}(k_i, k_j, k_r, m)}{4} + P_{ij}P_{adjacent}(k_i, k_j, m). \tag{31}$$

2. If $k_i > 1$ and $k_j > 1$,

$$
\begin{aligned}
E[W_{ij}] \quad &= \frac{P_{ij}}{4} \sum_{r \in V \setminus \{i,j\}} (P_{common\ neighbour}(k_i, k_j, k_r, m) - P_{triangle}(k_i, k_j, k_r, m)) \\
&\quad + 2P_{ij} \sum_{r \in V \setminus \{i,j\}} P_{triangle}(k_i, k_j, k_r, m) + 3P_{ij}P_{adjacency}(k_i, k_j, m),
\end{aligned} \tag{32}
$$

*Proof.* We can express the weights (4) depending on the cases as follows.
<u>If $k_i = 1$ or $k_j = 1$:</u>

$$W_{ij} = (1 - Y_{ij})\left(\frac{CN_{ij} - P_{ij}}{4}\right) + Y_{ij}P_{ij} = (1 - Y_{ij})\left(\frac{P_{ij}\sum_{r \in V \setminus \{i,j\}} Y_{ir}Y_{jr}}{4}\right) + Y_{ij}P_{ij}$$

Observe that if the term $\sum_{r \in V \setminus \{i,j\}} Y_{ir}Y_{jr} = 0$ then since the adjacency degree of $i$ or $j$ is one, $i$ and $j$ must be connected and therefore $Y_{ij} = 1$. Thus, the expression above results in $Y_{ij}P_{ij}$. Otherwise, if $\sum_{r \in V \setminus \{i,j\}} Y_{ir}Y_{jr} \neq 0$ again since the adjacency degree of $i$ or $j$ is one, $Y_{ij} = 0$ and the expression above simplifies to $\frac{P_{ij}\sum_{r \in V \setminus \{i,j\}} Y_{ir}Y_{jr}}{4}$. Hence, we obtain that

$$W_{ij} = \left(\frac{P_{ij}\sum_{r \in V \setminus \{i,j\}} Y_{ir}Y_{jr}}{4}\right) + Y_{ij}P_{ij}.$$

Next, we compute the expected values of the previous expression:

$$
\begin{aligned}
E[W_{ij}] \quad &= \left(\frac{P_{ij}\sum_{r \in V \setminus \{i,j\}} E[Y_{ir}Y_{jr}]}{4}\right) + P_{ij}E[Y_{ij}] \\
&= \left(\frac{P_{ij}\sum_{r \in V \setminus \{i,j\}} P(\{i \text{ and } r \text{ connected}\} \cap \{j \text{ and } r \text{ connected}\})}{4}\right) + P_{ij}P(\{i \text{ and } j \text{ connected}\}),
\end{aligned} \tag{33}
$$

and the result follows because the expression above coincides with (31).
<u>If $k_i > 1$ and $k_j > 1$:</u>

$$
\begin{aligned}
W_{ij} \quad &= (1 - Y_{ij})\left(\frac{CN_{ij} - P_{ij}}{4}\right) + Y_{ij}(2CN_{ij} + P_{ij}) \\
&= (1 - Y_{ij})\left(\frac{P_{ij}\sum_{r \in V \setminus \{i,j\}} Y_{ir}Y_{jr}}{4}\right) + Y_{ij}\left(2P_{ij}\left(\sum_{r \in V \setminus \{i,j\}} Y_{ir}Y_{jr} + 1\right) + P_{ij}\right).
\end{aligned}
$$

Then, the expected value of the expression above is:

$$
\begin{aligned}
E[W_{ij}] \quad &= \frac{P_{ij}\sum_{r\in V\setminus\{i,j\}}E[(1-Y_{ij})Y_{ir}Y_{jr}]}{4} + 2P_{ij}\sum_{r\in V\setminus\{i,j\}}E[Y_{ij}Y_{ir}Y_{jr}] + 3P_{ij}E[Y_{ij}] \\[2mm]
&= \frac{P_{ij}}{4}\sum_{r\in V\setminus\{i,j\}}\Big(P(\{i \text{ and } r \text{ connected}\} \cap \{j \text{ and } r \text{ connected}\}) \\[2mm]
&\quad -P(\{i \text{ and } r \text{ connected}\} \cap \{j \text{ and } r \text{ connected}\} \cap \{i \text{ and } j \text{ connected}\})\Big) \\[2mm]
&\quad +2P_{ij}\sum_{r\in V\setminus\{i,j\}}P(\{i \text{ and } r \text{ connected}\} \cap \{j \text{ and } r \text{ connected}\} \cap \{i \text{ and } j \text{ connected}\}) \\[2mm]
&\quad +3P_{ij}P(\{i \text{ and } j \text{ connected}\}).
\end{aligned}
\tag{34}
$$

Now, we observe that

$$
\begin{aligned}
P(i \text{ and } j \text{ connected}) &= P_{\text{adjacency}}(k_i, k_j, m) \\
P(i \text{ and } r \text{ connected}, j \text{ and } r \text{ connected}) &= P_{\text{common neighbour}}(k_i, k_j, k_r, m) \\
P(i \text{ and } r \text{ connected}, j \text{ and } r \text{ connected}, i \text{ and } j \text{ connected})) &= P_{\text{triangle}}(k_i, k_j, k_r, m)
\end{aligned}
$$

Finally, substituting the probabilities that appear in (33) and (34) with the expressions in (22), (23) and (24), one obtains the result.

**New models for detecting communities using weighted graph modularity games.** In the previous section, we show that the optimal solution of the analyzed instances provided by formulation $F_{Sh-JK}$ was the grand coalition except one node. Since, this type of solutions are meaningless for detecting overlapping communities, in this section, we provide an alternative model taking advantage of Theorem 1. Actually, we propose to define another modularity game $(N, \varphi)$, in which the characteristic function $\varphi$ is as in (1), but weights are defined as:

$$
W_{ij}^* = W_{ij} - W_{ij}^e,
\tag{35}
$$

where $W_{ij}^e = E(W_{ij})$. Observe that, the game is non-convex as $W_{ij}^*$ can take both positive and negative values.

To calculate the overlapping communities through the coalition stability of a modularity game, the objective function of formulation $F_{Sh-JK}$ must be modified according to Eq (35). Moreover, to avoid double counting (induced by pair of nodes that belongs to the same community in the new objective function), for any $1 \leq i < j \leq n$ the next binary variables are introduced:

$$
y_{ij} = \begin{cases} 1, & \text{if nodes } i \text{ and } j \text{ belong, at least once, to a common community,} \\[4mm] 0, & \text{otherwise.} \end{cases}
$$

Observe that if we would have used $y$-variables in model $F_{Sh-JK}$, the same solution would have been obtained because all the weights are positive and again the grand coalition would have been the optimal solution.

The final formulation of this model is:

$$(F^*_{Sh-Mod})\max \sum_{\substack{i,j=1 \\ i<j}}^{n} W^*_{ij} y_{ij} \tag{36}$$

s.t.: (5)–(7), (10), (13), (17), (18)

$$\sum_{\substack{j=1 \\ j\neq i}}^{n} W^*_{ij} x_{jk} \geq x_{ik} \frac{\sum_{\substack{j=1 \\ j\neq i}}^{n} W^*_{ij}}{2} + (1-x_{ik}) \sum_{\substack{j=1 \\ j\neq i \\ W^*_{ij}<0}}^{n} W^*_{ij}, \forall i=1,\ldots,n, k=1,\ldots,n_c, \tag{37}$$

$$\sum_{k=1}^{n_c} x_{ik} \leq p, \quad \forall i=1,\ldots,n, \tag{38}$$

$$y_{ij} \geq x_{ik} + x_{jk} - 1, \quad \forall i,j=1,\ldots,n,\, i<j,\, k=1,\ldots,n_c, \tag{39}$$

$$y_{ij} \leq \sum_{k=1}^{n_c} z_{ijk}, \quad \forall i,j=1,\ldots,n,\, i<j, \tag{40}$$

$$y_{ij} \in [0,1], \quad \forall i,j=1,\ldots,n,\, i<j. \tag{41}$$

The objective function (36) sums the weights between nodes of the same community only once. In this way, it cannot be the case that a community is a proper subset of another, because its profit would be null. Then, constraints (9), (14), (15), (16) and (19) that were discussed previously are not necessary. With (37) we guarantee that communities are stable for the new weights $W^*$. If $x_{ik}=1$, then (37) is equivalent to (8). Constraints (38) impose that each node cannot belong to more than $p$ different communities, with $p$ a fixed parameter established by the user. Constraints (39) and (40) impose that $y_{ij}=1$ if and only if there is a community $k$ to which $i$ and $j$ belong to. Finally, constraints (41) defines our variables as binary, but, from the arithmetic of the model, we can relax them as continuous variables ($y_{ij} \in [0,1]$) because in any case they can take only 0,1 values. The notation $F^*_{Sh-Mod}$ stands for the fact that the condition of stability is determined by the Shapley value of a modularity game with weights $W^*_{ij}$. In some experimental cases, it is interesting to compare the contribution of Theorem 1 over the approximations $W'_{ij}$, see (21), and therefore, we will refer as $F'_{Sh-Mod}$ to the model in which $W^*_{ij}$ are replaced by $W'_{ij}$.

The following experiments will highlight differences between models $F^*_{Sh-Mod}$ and $F'_{Sh-Mod}$, and differences between overlapping and non-overlapping communities models. The experiments are run in the Python environment and using the Gurobi solver.

In the first two examples we will show that models $F^*_{Sh-Mod}$, e.g. the exact model, and models $F'_{Sh-Mod}$, e.g. the approximation, compute different communities, even though they are run with the same parameters and the network size is small. From the tests, we can argue that the contribution of Theorem 1 is substantial.

We apply models $F^*_{Sh-Mod}$ and $F'_{Sh-Mod}$ to the Zachary's karate club network, [24], and compare the results with what obtained in [15]. The overlapping communities of that paper are three, so we fix $n_c=3$ and $p=2$. In Fig 3, each community is represented by the color grey, black or blue and the intersection nodes by red.

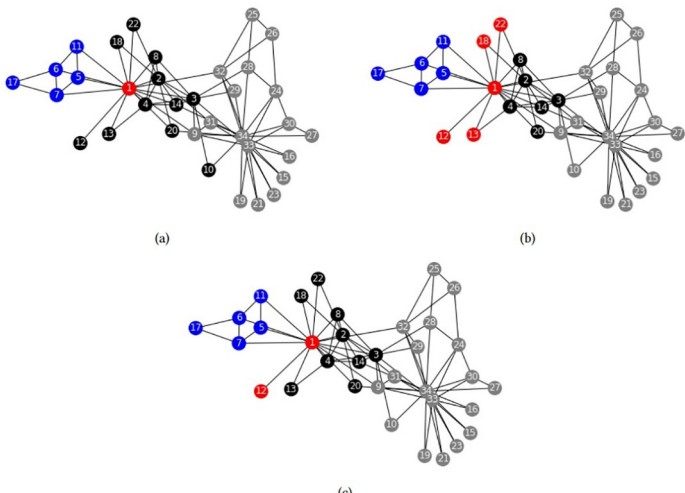

**Fig 3. Zachary's karate club community structures.** Community structures obtained by (a) [15], (b) $F^*_{Sh-Mod}$ with parameters $n_c = 3$, $p = 2$, (c) $F'_{Sh-Mod}$ with parameters $n_c = 3$, $p = 2$.

Fig 3a and 3c are similar. The only difference is that model $F'_{Sh-Mod}$ detects the node 12 as an intersection. It is reasonable, because node 12 is only connected to the other intersection node and share neighbours with both communities, black and blue. The structure obtained by model $F^*_{Sh-Mod}$ is also similar, but detects more intersection nodes, having connections with different communities and sharing neighbours with them. The results highlights that there can be differences between the exact and the approximate models, already when applied to small size graphs.

Next, we analyze models $F^*_{Sh-Mod}$ and $F'_{Sh-Mod}$ with other parameters. First, we fix $p = 1$, so that communities cannot overlap, and we obtain the results in Fig 4.

As can be seen, in both cases nodes that belong to the same community have high edge density between them and many common neighbours, even though the two communities in Fig 4a can be further split, as seen in Fig 4b. There, communities have higher edge density, but less common neighbors. It highlights the fact that equation (4) combines two criteria, namely density of common neighbors and number of connections, and the researcher must consider a trade-off between them. Letting communities overlap partially avoids this trade-off: With parameters $n_c = 4$ and $p = 2$, we obtain the results in Fig 5.

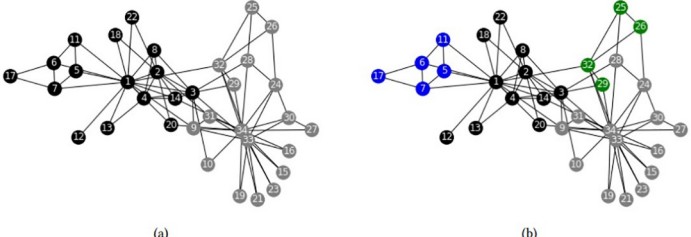

**Fig 4. Zachary's karate club disjoint community structures.** Community structures obtained by (a) $F^*_{Sh-Mod}$ with parameters $n_c = n$, $p = 1$, (b) $F'_{Sh-Mod}$ with parameters $n_c = n$, $p = 1$.

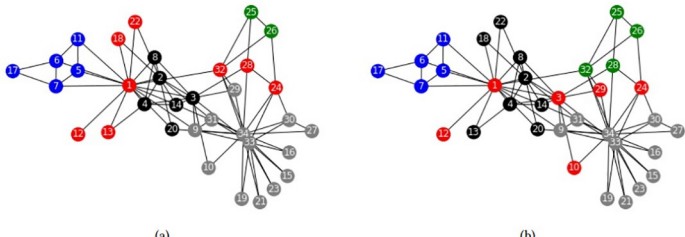

**Fig 5. Zachary's karate club community structures.** Community structures obtained by (a) $F^*_{Sh-Mod}$ with parameters $n_c = 4$, $p = 2$, (b) $F'_{Sh-Mod}$ with parameters $n_c = 4$, $p = 2$.

Figs 3b and 5a are similar. The intersection nodes found previously (Fig 3b) are also intersection nodes in Fig 5a with the new parameters. Nevertheless, some other intersection nodes appear that are brought about by the new fourth community of the clustering. Note that communities in Fig 5a are quite different from the ones of Fig 5b, especially for what concerns intersection nodes. As was remarked before, it implies that the differences between the exact and the approximate model are substantial.

Next, we apply models $F^*_{Sh-Mod}$ and $F'_{Sh-Mod}$ to the zebra communication network, see [25]. First, model $F^*_{Sh-Mod}$ is run with $p = 1$ and results are in Fig 6a. Results of model $F'_{Sh-Mod}$ are the same. Results of models $F^*_{Sh-Mod}$ and $F'_{Sh-Mod}$ with parameters $p = 2$ and $n_c = 3$ are in Fig 6b and 6c respectively. The former model does not detect any overlapping community, suggesting that they are well separated, while the latter model identifies node 20 as belonging to two communities. Since this model is actually an approximation of the real data, it is likely that the role of node 20 has been mistaken since the communities seems to be separated.

The following two examples compare the communities found by model $F^*_{Sh-Mod}$ when community i) cannot overlap ($p = 1$); ii) can overlap ($p > 1$). It will be seen that allowing

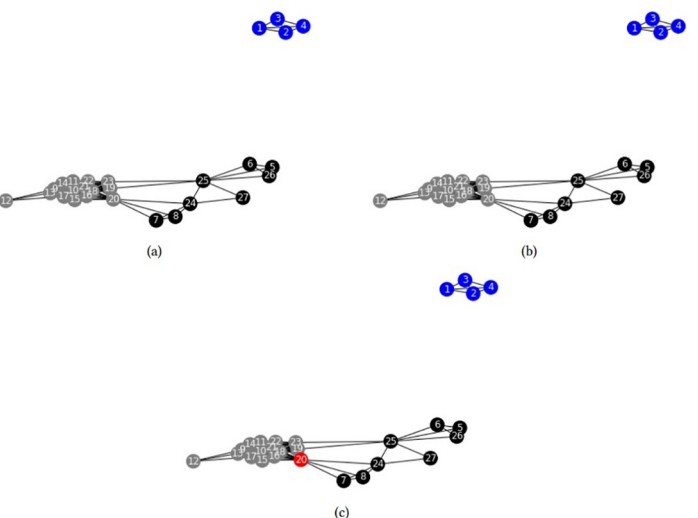

**Fig 6. Zebra community structures.** Community structures obtained by (a) $F^*_{Sh-Mod}$ with parameters $n_c = n$, $p = 1$, (b) $F^*_{Sh-Mod}$ with parameters $n_c = 3$, $p = 2$, (c) $F'_{Sh-Mod}$ with parameters $n_c = 3$, $p = 2$.

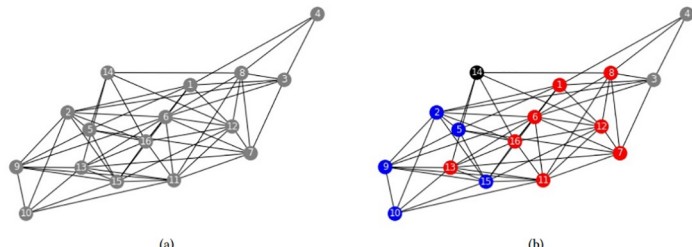

**Fig 7. Highland tribes community structures.** Community structures obtained by (a) $F^*_{Sh-Mod}$ with parameters $n_c = n$, $p = 1$, (b) $F^*_{Sh-Mod}$ with parameters $n_c = 3$, $p = 2$.

overlapping communities reveals nodes that are structurally different from others, forming the bulk of a core/periphery separation.

First, we apply the model $F^*_{Sh-Mod}$ to the the Highland tribes network, see [26]. First, model $F^*_{Sh-Mod}$ is run with $p = 1$ and results are in Fig 7a. There, it can be seen that, if no overlapping communities are allowed, then the model detects one community composed of all the nodes. Conversely, model $F^*_{Sh-Mod}$ is run with parameters $n_c = 3$ and $p = 2$, results are reported in Fig 7b. It can be seen that the role of different nodes is emerged. There, three communities of different size have been detected, with some nodes (the red ones) belonging to more than one community forming the core of the system of alliances.

Next, we apply model $F^*_{Sh-Mod}$ to the Windsurfers network, see [27]. Run with parameter $p = 1$, the model detected the two communities reported in Fig 8a. Run with parameters $n_c = 2$ and $p = 2$, the model detected the communities reported in Fig 8b. As can be seen, the results with overlapping communities are a refinement of the disjoint communities. Nodes that are in the border between the two groups are highlighted as members of both, forming the bulk of a core/periphery network segmentation.

To summarize our findings, the test of models $F^*_{Sh-Mod}$ on four typical benchmark networks revealed:

- Results between $F^*_{Sh-Mod}$ and $F'_{Sh-Mod}$ are different. As the latter is an approximation of the former, it reveals that the contribution of Theorem 1 to model development is substantial.

- Results between non-overlapping and overlapping community models are different. The former can reveal not only group membership, but nodes that could act as potential bridges between communities.

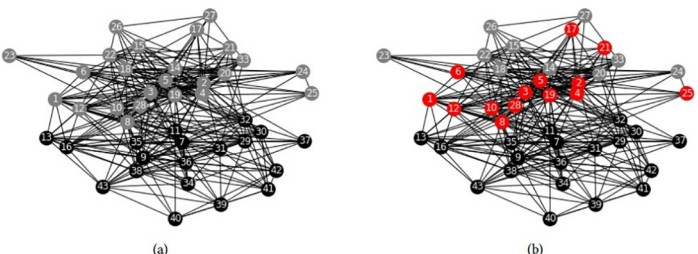

**Fig 8. Windsurfers community structures.** Community structures obtained by (a) $F^*_{Sh-Mod}$ with parameters $n_c = n$, $p = 1$, (b) $F^*_{Sh-Mod}$ with parameters $n_c = 2$, $p = 2$.

## Local Stability Exploration: An heuristic algorithms to detect overlapping communities

Problems $F^*_{Sh-Mod}$ and $F'_{Sh-Mod}$ are Integer Linear Programming (ILP) models whose solution computational times can be impractical when the instances to solve are large. This is normal when we deal with a NP-hard problem as the case of communities detection. Nevertheless, for large instances the ILP formulation can be applied to devise heuristic algorithms that could approximate the optimal solution in short computing time. Here we propose a method, that we will call Local Stability Exploration (LSE), that is based on local search. Suppose that a set of feasible communities $\Pi = \{S_1, \ldots, S_{n_c}\}, S_i \subseteq V, i = 1, \ldots, n_c$ is given, we will call such $\Pi$ an incumbent solution. $\Pi$ feasible means that it satisfies the ILP model constraints, so that i) every node belongs to at least one community, $\bigcup_{k=1}^{n_c} S_k = V$, ii) there is not strict inclusion between communities, $\nexists k, r = 1, \ldots, n_c, k \neq r$, such that $S_k \subseteq S_r$, iii) the maximum number of communities to which a node can belong is not exceeded by any node, i.e. $\forall i \in V$ the inequality $|\{k = 1, \ldots, n_c: i \in S_k\}| \leq p$ is fulfilled; and iv) all communities are stable. Next, we try to modify $\Pi$ to obtain a new feasible solution $\Pi'$ with an improved objective function. We consider three possible modification of $\Pi$, obtained by moves that are called Add, Remove, and Swap. Add is the move that joins a node to a community, allowing in this way multiple communities assignments. Remove is the move that takes away a node from a community. Swap is the move that switch two nodes between two communities. These moves are applied if and only if the new obtained $\Pi'$ is feasible. That is, after a move it must not occur that 1) a node does not belong to any community 2) a node belongs to more communities than allowed, maximum number of communities $p$ to which a node can belong; 3) one community is included in another, 4) modified communities are not stable.

For a feasible starting solution, the procedure is summarized in Algorithm 1. There, the triplet $(i, k, 1)$ is the move of adding node $i$ to community $k$, the triplet $(i, k, 2)$ is the move of removing node $i$ from community $k$, the 5-tuple $(i, k, i', k', 3)$ is swapping nodes $i$ and $i'$ between communities $k$ and $k'$. It can be seen that from Line 9 to Line 22 all feasible moves are considered. In Lines 12, 15 and 20 the increases of the objective function are calculated using the following notation: Let $C_i = \{k \in \{1, \ldots, n_c\}: i \in S_k\}$, that is, $C_i$ is the index set of the communities to which $i$ belongs, then the objective function can be written as:

$$f^*(\Pi) = \sum_{\substack{i,j \in V \\ i < j \\ C_i \cap C_j \neq \emptyset}}^{n} W^*_{ij}$$

Note that the condition $C_i \cap C_j \neq \emptyset$ is the condition that there is at least one community to which both $i$ and $j$ belong to. However, from the computational efficiency it is better to calculate just the increase of the objective function, as is done in lines 12, 15, 20. The new solution $\Pi'$ is the one that obtains the maximum increase. The algorithm stops when condition of Line 42 applies, as there are no improvements and a local optimum has been reached.

**Algorithm 1** Local stability exploration algorithm

```
1: procedure LOCAL STABILITY EXPLORATION
2:    Π = {S₁,...,Sₙ_c} ← Initial_Stable_Communities      ▷ Π is obtained by peculiar
                                                              subroutines
3:    for i in V do
4:       Cᵢ = {k ∈ {1, ..., n_c}: i ∈ S_k}
5:    end for
6:    f ← ∑ W*ᵢⱼ                                           ▷ Objective function
          i,j ∈ V
          i < j
          Cᵢ∩Cⱼ ≠ ∅
```

```
 7:    local_opt = FALSE                          ▷ Condition for a local optimum
 8:  while local_opt = FALSE do
 9:     Δ ← Feasible_Moves(Π)          ▷ Δ: list of admissible moves for Π.
10:      for (i, k, d) in Δ do
11:        if d = 1 then
```

$$12: \quad \delta_{ikd} \leftarrow \sum_{\substack{j \in S_k \\ C_i \cap C_j = \emptyset}} W_{ij}^*$$

```
13:        end if
14:        if d = 2 then
```

$$15: \quad \delta_{ikd} \leftarrow - \sum_{\substack{j \in S_k \setminus \{i\} \\ |C_i \cap C_j| = 1}} W_{ij}^*$$

```
16:        end if
18:      end for
18:      for (i, k, i', k', 3) ∈ Δ do
19:        if d = 3 then
```

$$20: \quad \delta_{iki'k'd} \leftarrow \sum_{\substack{j \in S_{k'} \setminus \{i'\} \\ C_i \cap C_j = \emptyset}} W_{ij}^* - \sum_{\substack{j \in S_k \setminus (S_{k'} \cup \{i\}) \\ |C_i \cap C_j| = 1}} W_{ij}^* + \sum_{\substack{j \in S_k \setminus \{i\} \\ C_{i'} \cap C_j = \emptyset}} W_{i'j}^* - \sum_{\substack{j \in S_{k'} \setminus (S_k \cup \{i'\}) \\ |C_{i'} \cap C_j| = 1}} W_{i'j}^*$$

```
21:        end if
22:      end for
23:     (i*, k*, d*) ∈ argmax{δ_{ikd}|(i, k, d)∈Δ}     ▷ Select the move that
                                                           increases the most
24:     (i*, k*, i'*, k'*, d*)∈argmax{δ_{iki'k'd}|(i, k, i', k', d)∈Δ}  ▷ Select
                                                      the move that increases the most
25:     if δ_{i*k*i'*k'*d*} > max{0, δ_{i*k*d*}} then
26:        f ← f+ δ_{i*k*i'*k'*d*}                                      ▷ Update f
27:        S_{k*} ← S_{k*} ∪ {i'*}\{i*}
28:        S_{k'*} ← S_{k'*} ∪ {i*}\{i'*}                              ▷ Update Π
29:        C_{i*} ← C_{i*} ∪ {k'*}\{k*}
30:        C_{i'*} ← C_{i'*} ∪ {k*}\{k'*}
31:     else
32:         if δ_{i*k*d*} > 0 then
33:          f ← f + δ_{i*k*d*}                                        ▷ Update f
34:          if d* = 1 then
35:            S_{k*} ← S_{k*} ∪ {i*}                                  ▷ Update Π
36:            C_{i*} ← C_{i*} ∪ {k*}
37:          else
38:            S_{k*} ← S_{k*}\{i*}                                     ▷ Update
39:            C_{i*} ← C_{i*}\{k*}
40:          end if
41:         else
42:          local_opt = TRUE
43:         end if
44:      end if
45:    end while
46:    return Π                                      ▷ Return the local optimum
47: end procedure
```

It remains to comment how feasible starting solutions can be obtained in Line 2 of Algorithm LSE. Depending on problems, we tested various procedures. The first possibility is to start with an unfeasible solution $\Pi$, because it contains unstable communities. Then Algorithm LSE is run without imposing that new solutions $\Pi'$ should be stable, but once that a feasible one has been found, then all forthcoming solutions must remain feasible too. The first unfeasible $\Pi$ can be a random assignment to communities, but another possibility is solving $F_{Sh-Mod}^*$ for $p = 1$, that is, when overlapping is not allowed, as the problem is usually solved faster than the cases in which $p > 1$. Another possibility that has been used for the problems with the

**Table 1. Computational results of the solution methods.**

| Dataset | $n_c$ | $p$ | Model | Solving method | Time (s) | Objective value |
|---|---|---|---|---|---|---|
| Zachary's karate club | 4 | 2 | $F^*_{Sh-Mod}$ | Exact ILP | 1530 | 162.469 |
| | | | | LSE heuristic | 77 | 162.469 |
| Zachary's karate club | 4 | 2 | $F'_{Sh-Mod}$ | Exact ILP | 316 | 129.39 |
| | | | | LSE heuristic | 6 | 129.279 |
| Zachary's karate club | 3 | 2 | $F^*_{Sh-Mod}$ | Exact ILP | 139 | 157.652 |
| | | | | LSE heuristic | 15 | 157.652 |
| Zachary's karate club | 3 | 2 | $F'_{Sh-Mod}$ | Exact ILP | 53 | 122.578 |
| | | | | LSE heuristic | 6 | 122.578 |
| Highland tribes | 3 | 2 | $F^*_{Sh-Mod}$ | Exact ILP | 31 | 89.654 |
| | | | | LSE heuristic | 4 | 89.654 |
| Highland tribes | 3 | 2 | $F'_{Sh-Mod}$ | Exact ILP | 8 | 47.4516 |
| | | | | LSE heuristic | 0.46 | 47.4516 |
| Zebra communication | 3 | 2 | $F^*_{Sh-Mod}$ | Exact ILP | 12 | 313.869 |
| | | | | LSE heuristic | 2.8 | 313.869 |
| Zebra communication | 3 | 2 | $F'_{Sh-Mod}$ | Exact ILP | 16 | 146.858 |
| | | | | LSE heuristic | 2.2 | 146.858 |
| Windsurfers | 2 | 2 | $F^*_{Sh-Mod}$ | Exact ILP | 1059 | 583.388 |
| | | | | LSE heuristic | 78 | 583.388 |
| Windsurfers | 2 | 2 | $F'_{Sh-Mod}$ | Exact ILP | 14 | 292.662 |
| | | | | LSE heuristic | 7.5 | 292.662 |

largest size is solving $F^*_{Sh-Mod}$ by branch-and-bound, but stop the search when the first feasible solution has been found and next using it as the starting solution in Line 2. All methods can be combined using any multi-start strategy, that is, repeating Algorithm 1 many times with different starting solutions to obtain sufficient diversification and exploration of the solution space. Finally, Algorithm LSE has been explained to solve model $F^*_{Sh-Mod}$, but it can be applied to $F'_{Sh-Mod}$ with straightforward modifications.

A preliminary test of the quality of the LSE algorithm has been run on the previous networks. We run a multi-start version allowing $t_{max} = 10$ starting solutions each run. Results about computational times and solution quality for different parameters configurations are reported in Table 1. It can be seen that the LSE heuristic algorithm reduces the computing time significantly with respect to the ILP solution for both models $F^*_{Sh-Mod}$ and $F'_{Sh-Mod}$, while the optimal solution has been achieved in all the cases but one.

Moreover, we applied the LSE algorithm to some large-scale real data sets that are impractical for any ILP model, in order to test the scalability of our heuristic. The solved data sets are the American college football network with 115 nodes, see [28], the Jazz musician network with 198 nodes, see [29], and C. metabolic network with 453 nodes, see [30]. These real data sets examples are commonly used in literature. The exact expected weights $W^*_{ij}$ cannot be computed for graphs with a large number of edges, so we used the approximated expected weights $W'_{ij}$. We report the results of our methods in Figs 9–11.

## Results and discussion

We are going to analyze the main features of the ILP models $F^*_{Sh-Mod}$, $F'_{Sh-Mod}$ and the heuristic Algorithm 1 when they are applied to medium and large size networks, most precisely,

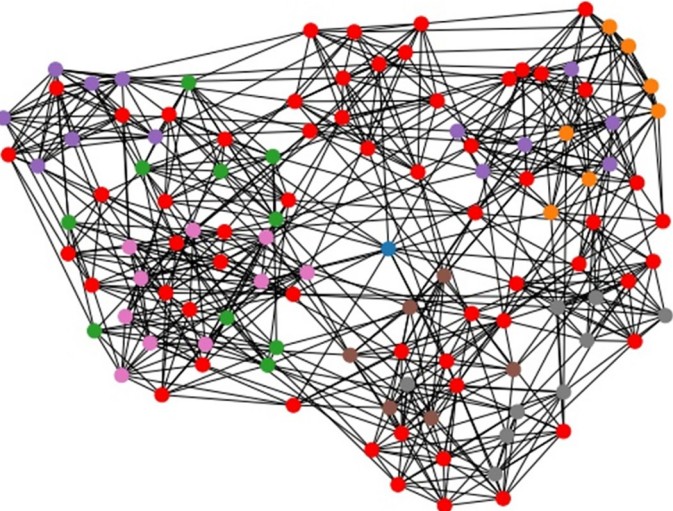

**Fig 9. American college football community structure.** Community structure obtained by LSE heuristic with $W'_{ij}$ weights and parameters $n_c = 7$, $p = 2$.

whether they can detect the true overlapping communities of randomly generated networks, as it is done in [31]. Random networks are generated using the procedure proposed in [16], but with some variations to allow for communities that overlap. Most peculiarly, in our simulation we must distinguish between bridge and non-bridge nodes, the former being the nodes that belongs to more than one community. The main parameters characterizing the simulated networks are:

- $N$: the number of nodes.

- $n_c$: the number of communities.

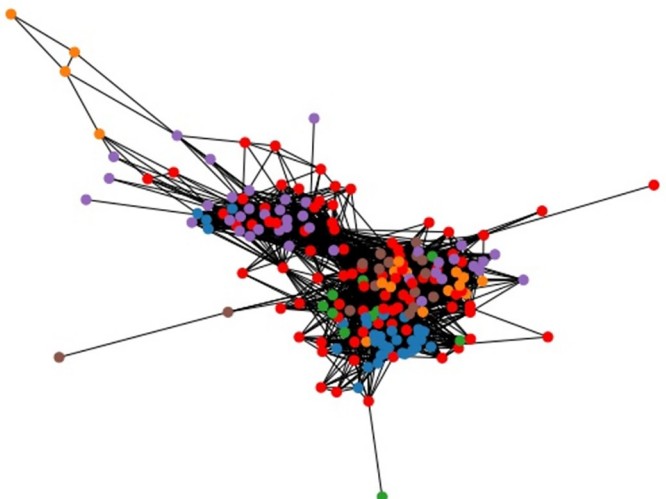

**Fig 10. Jazz music community structure.** Community structure obtained by LSE heuristic with $W'_{ij}$ weights and parameters $n_c = 6$, $p = 2$.

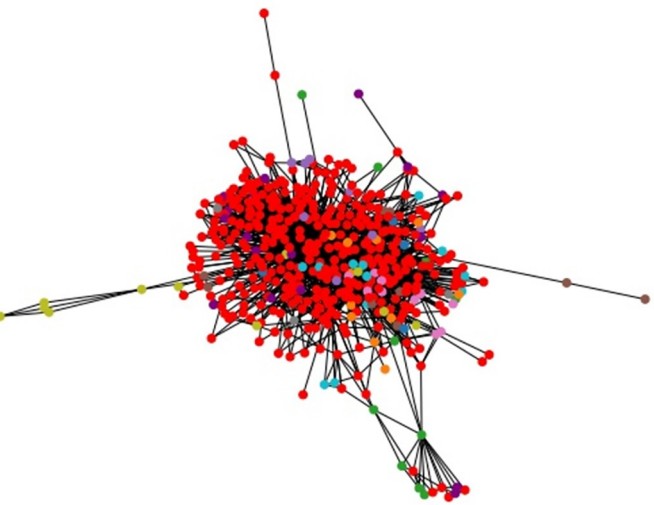

**Fig 11. C. metabolic community structure.** Community structure obtained by LSE heuristic with $W'_{ij}$ weights and parameters $n_c = 10$, $p = 2$.

- $p$: the maximum number of communities to which a node can belong to.

- $N_o$: the number of nodes that belongs to more than one communities, that is, they are bridges.

Next, communities are defined by the probability by which community nodes can establish a link between themselves. Those probabilities are controlled by parameters:

- $1 - \mu$: fraction of links between non-bridge nodes belonging to the same community.

- $1 - \mu_o$: fraction of links between bridge nodes and other nodes of the communities where the bridge node belongs to.

There are other parameters characterizing the simulated networks, such as the number of arcs, the node degrees, the community sizes and so on, whose purpose is to simulate networks with the same characteristics of the empiric ones. We report all these features in S1 Appendix, with the pseudo-code describing our implementation of Lancichenetti *et al.* algorithm.

The solution quality of our models is measured comparing their results with the true community structures (known by simulation). True and estimated structure may differ for:

- The community composition;

- The identification of the bridge nodes.

The statistics to compare the community composition are:

- the *Normalized Mutual Information (NMI) index* for overlapping partitions, presented in [32];

- the *Omega index (OI)*, presented in [33].

Both statistics range between 0 and 1, with values closer to 1 indicating strong correspondence between true and estimated communities.

The statistics to compare the identification of bridge nodes are based on a set of indices which depend on the values of the confusion matrix associated to the identification of bridge nodes. Each element of the confusion matrix is defined as follows

- True Positive (TP): Nodes successfully detected as bridge.

- True Negative (TN): Nodes successfully detected as non-bridge.

- False Positive (FP): Nodes wrongly detected as bridge.

- False Negative (FN): Nodes wrongly detected as non-bridge.

Then, we consider the following indices.

the *accuracy* defined as $\frac{TP+TN}{TP+TN+FP+FN}$,

the *True Positive Rate (TPR)*: $TPR = \frac{TP}{TP+FN}$,

the *False Positive Rate (FPR)*: $FPR = \frac{FP}{TN+FP}$,

the *Area Under Curve (AUC)*: $AUC = \frac{1-FPR+TPR}{2}$,

the *Precision* defined as $\frac{TP}{TP+FP}$,

the *F1 score*: $F1 = \frac{2TP}{2TP+FP+FN}$;

*Test 1: Detecting non overlapping communities:* As a first test, we apply the ILP models $F^*_{Sh-Mod}$, $F'_{Sh-Mod}$ and the Algorithms LSE to the case in which communities do not overlap, that is, $p = 1$, to see whether the approximate result of algorithm LSE are reliable, with respect to what is found by the respective optimal ILP models. The ILP solution of $F^*_{Sh-Mod}$ and $F'_{Sh-Mod}$ can be obtained in short computational times only for moderate size networks, so we consider $N = 40, 60$ to solve within the time limit of 100 or 200 seconds respectively. The LSE heuristic has been run with $t_{max} = 5$ multiple starting solution, guaranteeing that its computational times are a fraction of the exact method.

For fixed $N$ and $n_c$, we let $\mu = 0, 0.1, 0.2, 0.3, 0.4, 0.5, 0.6$, as in [16] to control for the effect of mixing parameter. For each parameter set, either 50 or 100 random networks are generated and indices are calculated as averages on all instances. Results are reported in Table 2. The first two rows of this table give the ILP formulation ($F^*_{Sh-Mod}$ or $F'_{Sh-Mod}$) used in the corresponding method: exact (ILP) or (LSE) heuristic to provide an initial solution. The third row describes the parameters of the instances ($N$, $n_c$, $\mu$) and the index reported below (NMI or Omega). By columns, the layout of this table is organized in three blocks. The first one with three columns describes the instances. The next two blocks, each one with four columns, report the average values of the NMI and Omega indices for each combination of solution method. Results in bold report the best behaviour among similar index for the corresponding solution methods. One can easily observe that using formulation $F^*_{Sh-Mod}$ in the ILP or in the LSE heuristic provides better solutions than $F'_{Sh-Mod}$.

For each combinations of parameters $N$ and $n_c$, the NMI and OI of each solution method are also shown as a function of $\mu$ in Figs 12–14 to compare the formulations $F^*_{Sh-Mod}$ and $F'_{Sh-Mod}$. The exact formulation $F^*_{Sh-Mod}$ obtains, in general, better NMI results and also better *OI* results in more cases than $F'_{Sh-Mod}$; except for $N = 40$ and $n_c = 4$. In this case, the behaviour of *OI* is similar in both formulations. However, also for $N = 40$, the exact solution of model $F^*_{Sh-Mod}$ is superior to the other two approaches, namely the heuristic LSE and the exact model $F'_{Sh-Mod}$, as the curves of the NMI and Omega statistics are above the others for most values of $N$, $n_c$ and $\mu$.

**Table 2. Computational results about networks with non-overlapping communities.**

| Model | | | $F^*_{Sh-Mod}$ | | | | $F'_{Sh-Mod}$ | | | |
|---|---|---|---|---|---|---|---|---|---|---|
| Method | | | ILP | | LSE | | ILP | | LSE | |
| N | $n_c$ | $\mu$ | NMI | OI | NMI | OI | NMI | OI | NMI | OI |
| 40 | 6 | 0 | **0.95** | 0.99 | **0.88** | **0.91** | **0.95** | **1** | 0.82 | 0.85 |
| | | 0.1 | **0.96** | 0.98 | **0.88** | **0.94** | 0.95 | **1** | 0.82 | 0.86 |
| | | 0.2 | **0.91** | 0.84 | **0.88** | **0.91** | 0.79 | **0.86** | 0.79 | 0.83 |
| | | 0.3 | **0.85** | **0.79** | **0.82** | **0.85** | 0.66 | 0.72 | 0.73 | 0.78 |
| | | 0.4 | **0.62** | 0.47 | **0.7** | **0.7** | 0.45 | **0.5** | 0.59 | 0.63 |
| | | 0.5 | **0.38** | **0.32** | **0.48** | **0.5** | 0.16 | 0.19 | 0.46 | 0.49 |
| | | 0.6 | **0.39** | **0.2** | **0.29** | **0.3** | 0.1 | 0.11 | 0.28 | **0.3** |
| 40 | 4 | 0 | **0.88** | 0.99 | **0.87** | **0.98** | **0.88** | **1** | 0.83 | 0.92 |
| | | 0.1 | **0.88** | 0.99 | **0.86** | **0.99** | **0.88** | **1** | 0.81 | 0.91 |
| | | 0.2 | **0.88** | 0.94 | **0.85** | **0.97** | **0.88** | **0.99** | 0.83 | 0.92 |
| | | 0.3 | **0.82** | 0.79 | **0.83** | **0.93** | 0.78 | **0.88** | 0.81 | 0.9 |
| | | 0.4 | **0.63** | 0.63 | **0.78** | **0.87** | 0.57 | **0.68** | 0.76 | 0.85 |
| | | 0.5 | **0.35** | **0.34** | **0.6** | **0.71** | 0.17 | 0.21 | 0.57 | 0.65 |
| | | 0.6 | **0.15** | **0.23** | **0.36** | **0.41** | 0.07 | 0.09 | 0.34 | **0.41** |
| 60 | 6 | 0 | **0.79** | 0.85 | **0.89** | **0.98** | **0.79** | **0.87** | 0.83 | 0.92 |
| | | 0.1 | **0.72** | **0.78** | **0.9** | **0.99** | 0.53 | 0.6 | 0.84 | 0.93 |
| | | 0.2 | **0.7** | **0.73** | **0.9** | **0.98** | 0.33 | 0.41 | 0.83 | 0.92 |
| | | 0.3 | **0.63** | **0.65** | **0.89** | **0.97** | 0.07 | 0.08 | 0.82 | 0.9 |
| | | 0.4 | **0.36** | **0.39** | **0.87** | **0.93** | 0 | 0.01 | 0.79 | 0.87 |
| | | 0.5 | **0.28** | **0.3** | **0.73** | **0.79** | 0 | 0 | 0.66 | 0.74 |
| | | 0.6 | **0.13** | **0.15** | **0.44** | **0.52** | 0 | 0 | 0.39 | 0.47 |

Maximum *NMI* and *OI* values for each combination of parameters and each method are highlighted in bold.

When $\mu$ is above the threshold 0.3, the solution quality of the method deteriorates for the joint effect of two factors: 1) communities are less well-separated, 2) exact solution has not been obtained within the considered time limit. However, this is not an actual drawback since for those parameter values, communities are essentially meaningless.

*Test 2: Detecting overlapping communities on small networks:* Networks with overlapping communities have been simulated with the same parameters used before, but now communities can overlap. We control the overlap with parameters $p = \{2, 3\}$ and $\mu_o = \{0.5, 0.7\}$. The choice of these parameters is justified since for $p = 2$ the smallest possible $\mu_o$ value is 0.5 and for $p = 3$ the smallest possible $\mu_o$ value is approximately 0.7. Moreover, the number of bridge

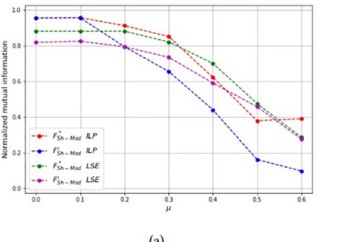
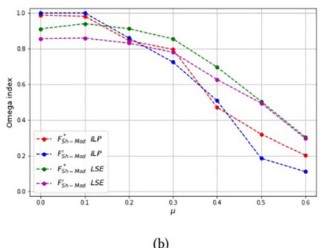

(a)                    (b)

**Fig 12. Test results on non-overlapping communities, parameters $N = 40$, $n_c = 6$.** (a) Average *NMI* for each solution method, (b) Average *Omega* for each solution method.

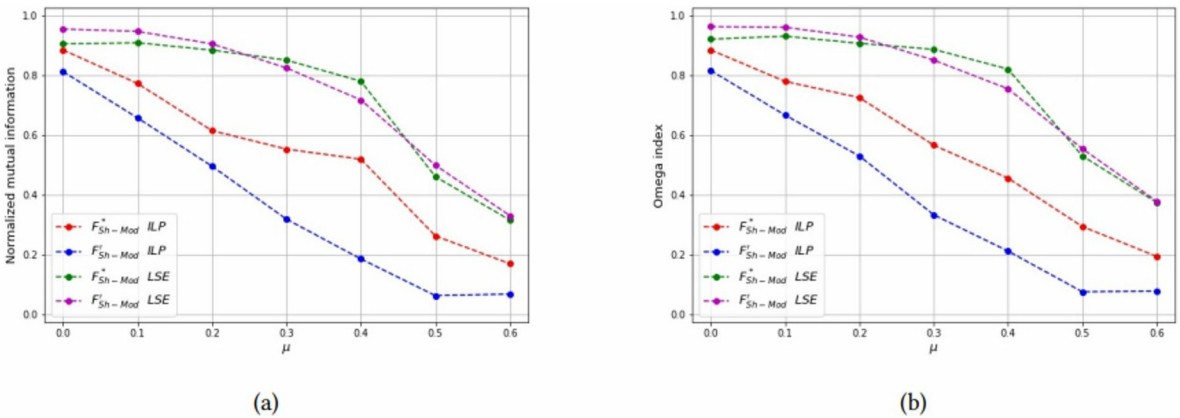

**Fig 13. Test results on non-overlapping communities, parameters $N = 40$, $n_c = 4$.** (a) Average *NMI* for each solution method, (b) Average *Omega* for each solution method.

nodes $N_o$ is approximately 10% of all the nodes, and we change this value to asses how it affects the computational results. Problems with overlapping communities are harder to solve, therefore we limit the graph size to $N = 40$ and increase the time limit to 200 seconds. Table 3 reports the computational results with a layout similar to Table 2. It can be seen that the best values of both the Omega and NME indices are obtained with the LSE heuristic, applied to the $F'_{Sh-Mod}$ formulation. The LSE heuristic applied to $F^*_{Sh-Mod}$ provides the second best results (with a few exceptions in which it becomes the best one) and the third one is the ILP formulation. The reason of the poor performance of the ILP methods is due to the fact that they were not able to terminate the computation in the imposed time limit and the solution that they provide is far from optimality. Results of Table 3 are graphically reported in Figs 15–19, where it can be seen that the purple and green curve, representing the LSE heuristics, are very close with each other and they are much above the result of the truncated ILP. It is also noteworthy that weights from $F^*_{Sh-Mod}$ improve the results of the ILP method.

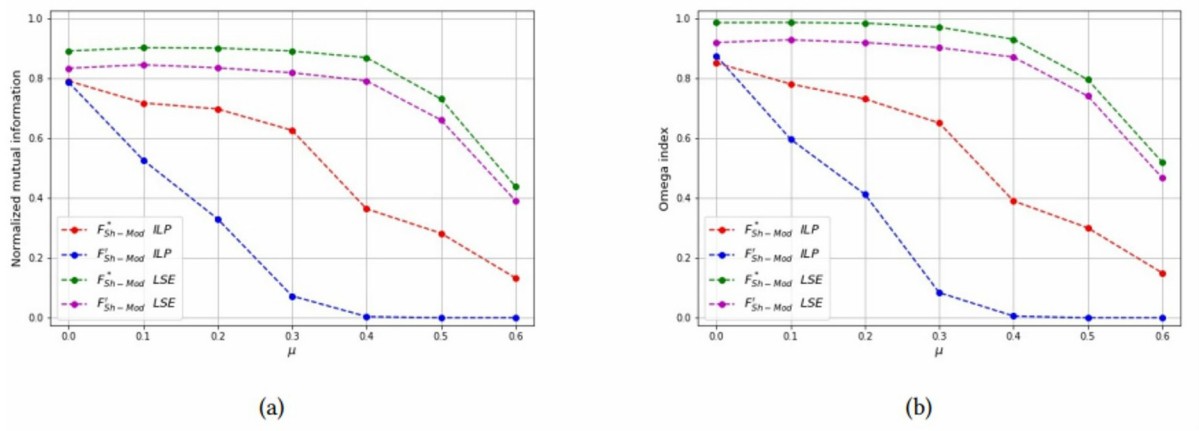

**Fig 14. Test results on non-overlapping communities, parameters $N = 60$, $n_c = 6$.** (a) Average *NMI* for each solution method, (b) Average *Omega* for each solution method.

**Table 3. Computational results about networks with overlapping communities.**

| Model | | | | | $F^*_{Sh-Mod}$ | | | | $F'_{Sh-Mod}$ | | | |
| --- | --- | --- | --- | --- | --- | --- | --- | --- | --- | --- | --- | --- |
| Method | | | | | ILP | | LSE | | ILP | | LSE | |
| $n_c$ | $p$ | $\mu_o$ | $N_o$ | $\mu$ | NMI | OI | NMI | OI | NMI | OI | NMI | OI |
| 4 | 2 | 0.5 | 1 | 0 | **0.88** | **0.92** | 0.86 | 0.92 | **0.88** | 0.87 | **0.91** | **0.95** |
| | | | | 0.1 | **0.76** | **0.78** | 0.88 | 0.91 | 0.71 | 0.73 | **0.9** | **0.95** |
| | | | | 0.2 | **0.63** | **0.66** | 0.85 | 0.91 | 0.45 | 0.46 | **0.88** | **0.92** |
| | | | | 0.3 | **0.57** | **0.63** | 0.82 | **0.88** | 0.28 | 0.3 | **0.83** | **0.88** |
| | | | | 0.4 | **0.4** | **0.46** | **0.76** | **0.81** | 0.17 | 0.18 | 0.72 | 0.78 |
| | | | | 0.5 | **0.28** | **0.33** | 0.49 | **0.57** | 0.1 | 0.12 | **0.5** | **0.57** |
| | | | | 0.6 | **0.13** | **0.16** | 0.29 | 0.35 | 0.05 | 0.05 | **0.31** | **0.37** |
| | | | 3 | 0 | **0.88** | **0.88** | 0.9 | 0.92 | 0.81 | 0.82 | **0.95** | **0.96** |
| | | | | 0.1 | **0.77** | **0.78** | 0.91 | 0.93 | 0.66 | 0.67 | **0.95** | **0.96** |
| | | | | 0.2 | **0.61** | **0.72** | 0.88 | 0.91 | 0.5 | 0.53 | **0.9** | **0.93** |
| | | | | 0.3 | **0.55** | **0.57** | **0.85** | **0.89** | 0.32 | 0.33 | 0.82 | 0.85 |
| | | | | 0.4 | **0.52** | **0.46** | **0.78** | **0.82** | 0.19 | 0.21 | 0.72 | 0.75 |
| | | | | 0.5 | **0.26** | **0.29** | 0.46 | 0.53 | 0.06 | 0.08 | **0.5** | **0.55** |
| | | | | 0.6 | **0.17** | **0.19** | 0.32 | **0.38** | 0.07 | 0.08 | **0.33** | **0.38** |
| | | | 5 | 0 | **0.74** | **0.79** | 0.91 | 0.93 | 0.72 | 0.71 | **0.94** | **0.95** |
| | | | | 0.1 | **0.67** | **0.69** | 0.9 | 0.91 | 0.66 | 0.67 | **0.96** | **0.97** |
| | | | | 0.2 | **0.58** | **0.65** | **0.91** | **0.92** | 0.42 | 0.46 | 0.9 | 0.91 |
| | | | | 0.3 | **0.53** | **0.56** | **0.85** | **0.87** | 0.28 | 0.31 | 0.82 | 0.85 |
| | | | | 0.4 | **0.42** | **0.42** | **0.7** | **0.76** | 0.15 | 0.17 | 0.64 | 0.68 |
| | | | | 0.5 | **0.28** | **0.27** | **0.44** | **0.52** | 0.06 | 0.06 | 0.44 | 0.49 |
| | | | | 0.6 | **0.15** | **0.21** | **0.31** | **0.37** | 0.06 | 0.07 | 0.28 | 0.32 |
| | | 0.7 | 3 | 0 | 0.87 | 0.8 | 0.88 | 0.9 | **0.94** | **0.94** | **0.94** | **0.96** |
| | | | | 0.1 | 0.77 | 0.68 | 0.88 | 0.9 | **0.79** | **0.8** | **0.95** | **0.96** |
| | | | | 0.2 | **0.71** | **0.7** | 0.88 | **0.91** | 0.46 | 0.48 | 0.89 | **0.91** |
| | | | | 0.3 | **0.55** | **0.54** | 0.83 | 0.87 | 0.24 | 0.26 | **0.85** | **0.88** |
| | | | | 0.4 | **0.42** | **0.38** | **0.72** | **0.77** | 0.18 | 0.21 | 0.67 | 0.71 |
| | | | | 0.5 | **0.23** | **0.29** | **0.49** | **0.56** | 0.08 | 0.09 | 0.47 | 0.51 |
| | | | | 0.6 | **0.17** | **0.15** | **0.33** | **0.39** | 0.08 | 0.09 | 0.29 | 0.34 |
| | 3 | 0.7 | 3 | 0 | 0.74 | 0.7 | 0.82 | 0.84 | **0.8** | **0.83** | **0.88** | **0.89** |
| | | | | 0.1 | **0.67** | **0.64** | 0.85 | 0.87 | 0.57 | 0.6 | **0.87** | **0.89** |
| | | | | 0.2 | **0.59** | **0.56** | **0.83** | **0.86** | 0.3 | 0.34 | **0.83** | 0.85 |
| | | | | 0.3 | **0.5** | **0.42** | **0.76** | **0.8** | 0.18 | 0.2 | 0.71 | 0.76 |
| | | | | 0.4 | **0.32** | **0.41** | **0.63** | **0.69** | 0.17 | 0.21 | 0.56 | 0.62 |
| | | | | 0.5 | **0.22** | **0.26** | 0.35 | 0.43 | 0.11 | 0.12 | **0.37** | **0.44** |
| | | | | 0.6 | **0.14** | **0.16** | 0.21 | **0.27** | 0.07 | 0.08 | **0.22** | 0.26 |

Maximum *NMI* and *OI* values for each combination of parameters and each method are highlighted in bold.

*Test 3: Detecting overlapping communities on large-scale networks:* In the last experiment, we have applied the LSE heuristics, using both the $F^*_{Sh-Mod}$ and $F'_{Sh-Mod}$ models, to the largest networks composed of 500 or 1000 nodes. As before, we control the overlap between communities with parameters $p = \{2, 3\}$ and $\mu_o = \{0.6, 0.7\}$, the number of bridge nodes are $N_o = \{20, 50\}$.

In Table 4, we report the *NMI* and *OI* statistics calculated by the two methods. It can be seen that they have lower values than what obtained in the smallest networks, due the fact that

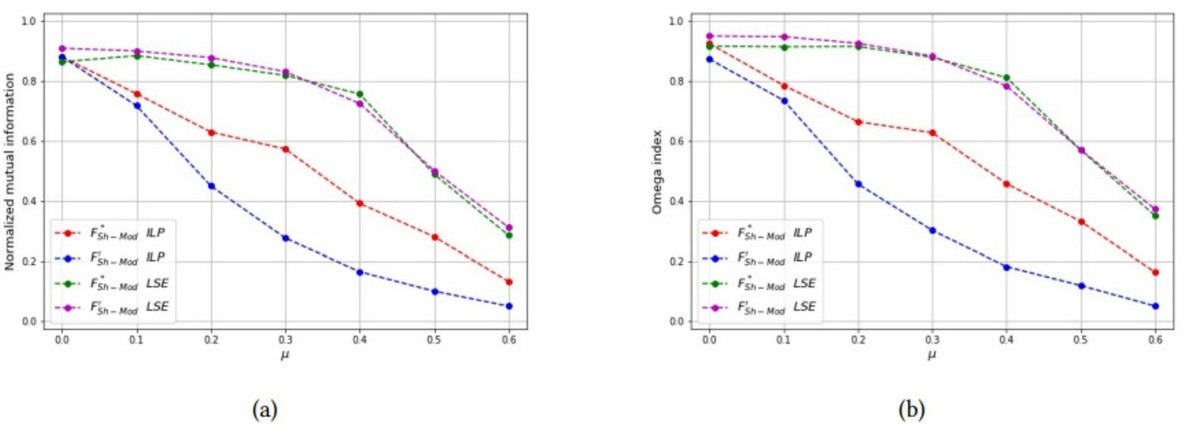

**Fig 15. Test results on overlapping communities, parameters $p = 2$, $\mu_o = 0.5$, $N_o = 1$.** (a) Average *NMI* for each solution method, (b) Average *Omega* for each solution method.

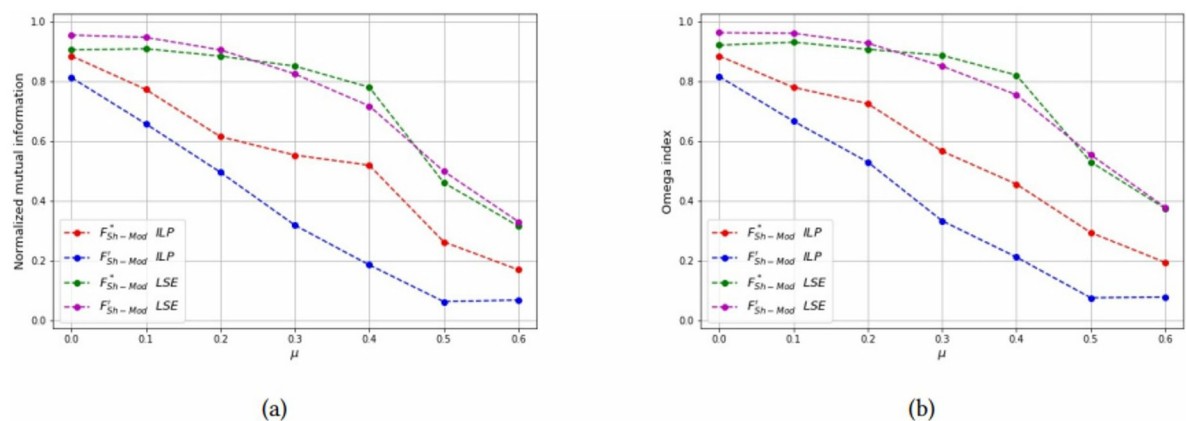

**Fig 16. Test results on overlapping communities, parameters $p = 2$, $\mu_o = 0.5$, $N_o = 3$.** (a) Average *NMI* for each solution method, (b) Average *Omega* for each solution method.

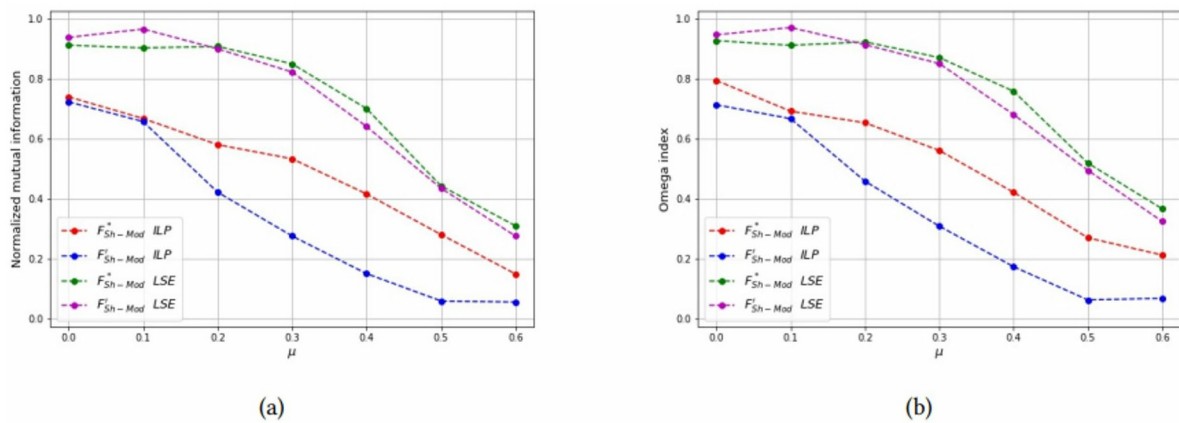

**Fig 17. Test results on overlapping communities, parameters $p = 2$, $\mu_o = 0.5$, $N_o = 5$.** (a) Average *NMI* for each solution method, (b) Average *Omega* for each solution method.

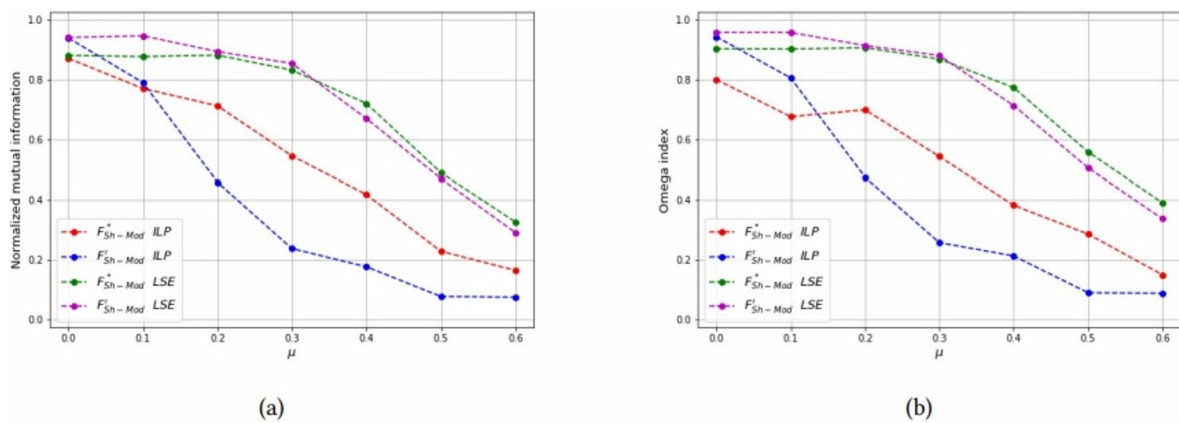

**Fig 18. Test results on overlapping communities, parameters $p = 2$, $\mu_o = 0.7$, $N_o = 3$.** (a) Average *NMI* for each solution method, (b) Average *Omega* for each solution method.

communities are harder to find. in most of the cases, model $F^*_{Sh-Mod}$, in which weights are exact, obtains better indices than the approximated weights of $F'_{Sh-Mod}$. Results of Table 4 are reported in Figs 20–23. There, it can be seen that the green line is above the purple one in almost all cases.

We can compare models $F^*_{Sh-Mod}$ and $F'_{Sh-Mod}$ in term of detecting the network bridge nodes. We considered many statistics: *accuracy*, *TPR*, *FPR*, *AUC*, *precision* and the *F1 score*. They are collected in Table 5 which reports the average values of these metrics obtained by the two LSE heuristics. In all the simulations, the fraction of bridge nodes over all the nodes is less than 0.1. It implies that it is much easier to detect non-bridge nodes rather than bridge ones. Therefore, a method that selects the fewest number of bridge nodes has a numeric advantage in terms of *accuracy*. Clearly, it could not classify successfully bridge nodes. Looking at Table 5, one can observe that the greatest difference between $F^*_{Sh-Mod}$-LSE and $F'_{Sh-Mod}$-LSE is on metrics *FPR* and *TPR*. Model $F^*_{Sh-Mod}$ obtains the best rate of *true positive*, model $F'_{Sh-Mod}$-LSE obtains the

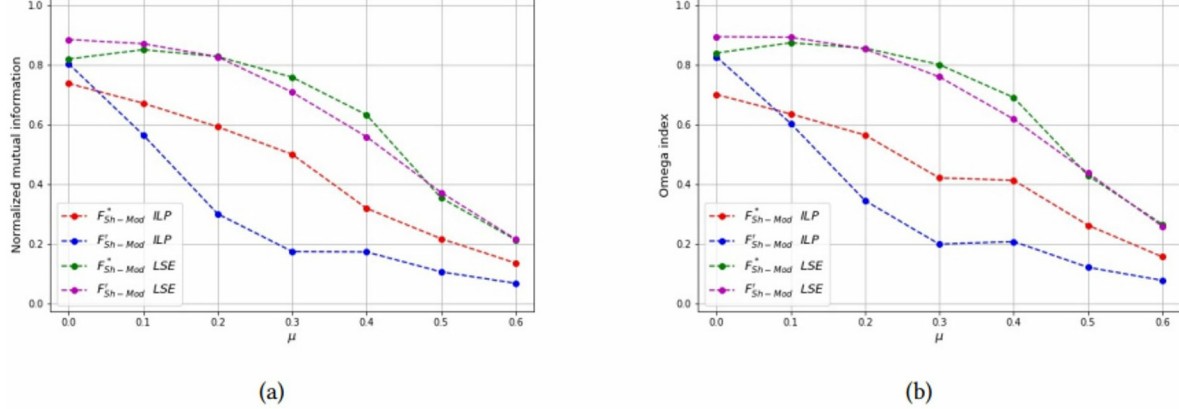

**Fig 19. Test results on overlapping communities, parameters $p = 3$, $\mu_o = 0.7$, $N_o = 3$.** (a) Average *NMI* for each solution method, (b) Average *Omega* for each solution method.

**Table 4. Computational results about large-scale networks with overlapping communities.**

| Model | | | | | | $F^*_{Sh-Mod}$ | | $F'_{Sh-Mod}$ | |
| --- | --- | --- | --- | --- | --- | --- | --- | --- | --- |
| Method | | | | | | LSE | | LSE | |
| $N$ | $n_c$ | $p$ | $\mu_o$ | $N_o$ | $\mu$ | NMI | Omega | NMI | Omega |
| 500 | 25 | 2 | 0.6 | 20 | 0 | **0.63** | **0.74** | 0.59 | 0.69 |
| | | | | | 0.1 | **0.61** | **0.73** | 0.58 | 0.69 |
| | | | | | 0.2 | **0.53** | **0.6** | 0.46 | 0.56 |
| | | | | | 0.3 | **0.45** | 0.51 | 0.43 | **0.53** |
| | | | | | 0.4 | **0.38** | **0.44** | 0.37 | 0.42 |
| | | | | | 0.5 | **0.33** | **0.36** | 0.32 | 0.33 |
| | | | | | 0.6 | 0.26 | **0.29** | **0.27** | 0.25 |
| | | | | 50 | 0 | **0.43** | **0.56** | 0.42 | 0.52 |
| | | | | | 0.1 | **0.46** | 0.57 | **0.46** | **0.58** |
| | | | | | 0.2 | **0.41** | 0.51 | **0.41** | **0.53** |
| | | | | | 0.3 | **0.39** | 0.46 | **0.39** | **0.5** |
| | | | | | 0.4 | **0.33** | **0.41** | **0.33** | 0.4 |
| | | | | | 0.5 | **0.3** | **0.35** | **0.3** | 0.33 |
| | | | | | 0.6 | 0.26 | **0.29** | **0.28** | 0.28 |
| | | 3 | 0.7 | 20 | 0 | **0.62** | 0.71 | 0.61 | **0.71** |
| | | | | | 0.1 | **0.62** | **0.73** | 0.53 | 0.64 |
| | | | | | 0.2 | **0.56** | **0.58** | 0.46 | 0.57 |
| | | | | | 0.3 | **0.5** | **0.59** | 0.4 | 0.48 |
| | | | | | 0.4 | **0.46** | **0.49** | 0.36 | 0.42 |
| | | | | | 0.5 | **0.38** | **0.36** | 0.33 | 0.34 |
| | | | | | 0.6 | **0.31** | **0.27** | 0.26 | 0.23 |
| 1000 | 50 | 2 | 0.6 | 50 | 0 | 0.41 | 0.55 | **0.6** | **0.7** |
| | | | | | 0.1 | 0.42 | 0.57 | **0.46** | **0.6** |
| | | | | | 0.2 | **0.44** | **0.58** | 0.38 | 0.5 |
| | | | | | 0.3 | **0.38** | **0.49** | 0.32 | 0.41 |
| | | | | | 0.4 | **0.35** | **0.41** | 0.23 | 0.32 |
| | | | | | 0.5 | **0.14** | **0.27** | 0.13 | 0.25 |
| | | | | | 0.6 | **0.09** | **0.22** | 0.06 | 0.19 |

Maximum *NMI* and *OI* values for each combination of parameters and each method are highlighted in bold.

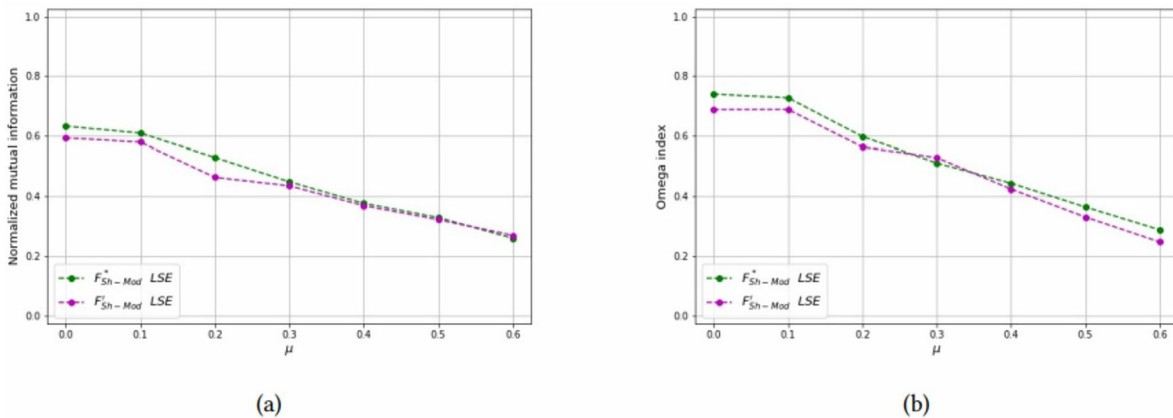

**Fig 20. Test results on overlapping communities, parameters $N = 500$, $n_c = 25$, $p = 2$, $\mu_o = 0.6$, $N_o = 20$.** (a) Average *NMI* for each solution method, (b) Average *Omega* for each solution method.

**Table 5. Computational results about large-scale networks with overlapping communities.**

| Model | | | | | | $F^*_{Sh-Mod}$ | | | | | | $F'_{Sh-Mod}$ | | | | | |
|---|---|---|---|---|---|---|---|---|---|---|---|---|---|---|---|---|---|
| Method | | | | | | LSE | | | | | | LSE | | | | | |
| N | $n_c$ | p | $\mu_o$ | $N_o$ | $\mu$ | Accuracy | TPR | FPR | AUC | Precision | F1 | Accuracy | TPR | FPR | AUC | Precision | F1 |
| 500 | 25 | 2 | 0.6 | 20 | 0 | 0.87 | **0.81** | 0.13 | **0.84** | **0.83** | **0.82** | **0.98** | 0.69 | **0** | **0.84** | 0.75 | 0.72 |
| | | | | | 0.1 | 0.86 | **0.82** | 0.14 | 0.84 | **0.69** | **0.75** | **0.95** | 0.74 | **0.04** | **0.85** | 0.51 | 0.6 |
| | | | | | 0.2 | 0.73 | **0.84** | 0.28 | 0.78 | **0.32** | **0.46** | **0.91** | 0.65 | **0.08** | **0.79** | 0.21 | 0.32 |
| | | | | | 0.3 | 0.59 | **0.87** | 0.42 | 0.72 | **0.15** | **0.26** | **0.83** | 0.73 | **0.16** | **0.78** | 0.14 | 0.23 |
| | | | | | 0.4 | 0.49 | **0.89** | 0.53 | 0.68 | **0.09** | **0.16** | **0.73** | 0.7 | **0.27** | **0.71** | 0.08 | 0.14 |
| | | | | | 0.5 | 0.38 | **0.88** | 0.64 | 0.62 | **0.06** | **0.11** | **0.58** | 0.69 | **0.42** | **0.63** | **0.06** | **0.11** |
| | | | | | 0.6 | 0.27 | **0.87** | 0.76 | **0.56** | **0.05** | **0.09** | **0.46** | 0.65 | **0.55** | 0.55 | 0.04 | 0.08 |
| | | | | 50 | 0 | 0.74 | **0.87** | 0.28 | 0.8 | **0.64** | **0.74** | **0.95** | 0.63 | **0.01** | **0.81** | 0.63 | 0.63 |
| | | | | | 0.1 | 0.76 | **0.86** | 0.26 | 0.8 | **0.66** | **0.75** | **0.94** | 0.73 | **0.03** | **0.85** | 0.63 | 0.68 |
| | | | | | 0.2 | 0.66 | **0.88** | 0.37 | 0.75 | **0.45** | **0.6** | **0.9** | 0.69 | **0.07** | **0.81** | 0.44 | 0.54 |
| | | | | | 0.3 | 0.6 | **0.89** | 0.43 | 0.73 | **0.32** | **0.47** | **0.83** | 0.73 | **0.16** | **0.79** | 0.3 | 0.43 |
| | | | | | 0.4 | 0.49 | **0.9** | 0.55 | 0.67 | **0.21** | **0.34** | **0.74** | 0.7 | **0.25** | **0.73** | 0.2 | 0.31 |
| | | | | | 0.5 | 0.42 | **0.89** | 0.63 | 0.63 | **0.16** | **0.27** | **0.6** | 0.7 | **0.4** | **0.65** | 0.14 | 0.23 |
| | | | | | 0.6 | 0.37 | **0.85** | 0.69 | **0.58** | **0.13** | **0.23** | **0.46** | 0.73 | **0.57** | **0.58** | 0.12 | 0.21 |
| | | 3 | 0.7 | 20 | 0 | 0.89 | **0.99** | 0.12 | 0.94 | 0.83 | **0.9** | **0.99** | 0.91 | **0** | **0.95** | **0.85** | 0.88 |
| | | | | | 0.1 | 0.93 | **0.99** | 0.08 | **0.96** | **0.77** | **0.87** | **0.97** | 0.83 | **0.02** | 0.91 | 0.57 | 0.68 |
| | | | | | 0.2 | 0.74 | **0.99** | 0.27 | 0.86 | **0.35** | **0.52** | **0.93** | 0.83 | **0.07** | **0.88** | 0.28 | 0.42 |
| | | | | | 0.3 | 0.78 | **0.99** | 0.22 | **0.88** | **0.23** | **0.37** | **0.85** | 0.79 | **0.15** | 0.82 | 0.15 | 0.25 |
| | | | | | 0.4 | 0.49 | **0.99** | 0.37 | **0.81** | **0.13** | **0.23** | **0.75** | 0.86 | **0.26** | 0.8 | 0.11 | 0.2 |
| | | | | | 0.5 | 0.46 | **0.99** | 0.56 | 0.71 | **0.08** | **0.15** | **0.6** | 0.94 | **0.41** | **0.76** | **0.08** | **0.15** |
| | | | | | 0.6 | 0.35 | **0.98** | 0.67 | 0.65 | **0.06** | **0.11** | **0.5** | 0.83 | **0.51** | **0.66** | 0.05 | 0.09 |
| 1000 | 50 | 2 | 0.6 | 50 | 0 | **0.98** | 0.66 | **0.01** | **0.83** | **0.8** | **0.72** | 0.96 | **0.82** | 0.03 | **0.9** | 0.6 | 0.69 |
| | | | | | 0.1 | **0.95** | **0.8** | **0.05** | **0.88** | **0.47** | **0.59** | 0.88 | **0.8** | 0.1 | 0.85 | 0.28 | 0.41 |
| | | | | | 0.2 | **0.89** | **0.82** | **0.11** | **0.86** | **0.29** | **0.43** | 0.79 | 0.81 | 0.2 | 0.81 | 0.17 | 0.28 |
| | | | | | 0.3 | **0.81** | 0.74 | **0.19** | **0.78** | **0.17** | **0.28** | 0.68 | **0.8** | 0.32 | 0.72 | 0.12 | 0.21 |
| | | | | | 0.4 | **0.62** | **0.78** | **0.39** | **0.7** | **0.1** | **0.18** | 0.54 | 0.76 | 0.47 | 0.65 | 0.08 | 0.14 |
| | | | | | 0.5 | **0.51** | 0.7 | **0.5** | **0.6** | **0.07** | **0.13** | 0.38 | **0.76** | 0.64 | 0.56 | 0.06 | 0.11 |
| | | | | | 0.6 | 0.25 | **0.75** | 0.82 | **0.47** | **0.07** | **0.13** | **0.26** | 0.73 | 0.77 | **0.48** | 0.05 | 0.09 |

Best values for each combinations of parameters are highlighted in bold.

best rate of *false positive*. This means that model $F^*_{Sh-Mod}$ selects more bridge nodes, but some of them are not actually bridges. Conversely, $F'_{Sh-Mod}$ can successfully detect most of the non-bridge nodes, resulting on higher *accuracy* just because the majority of nodes are actually non-bridge. However, this is a consequence of a method that takes less risk in detecting a node as a bridge. As far as the *AUC* is concerned, the results are really similar due to the existing balance between *FPR* and *TPR* of both methods.

These values confirm that the bridge nodes detected by model $F^*_{Sh-Mod}$ are more reliable than the ones detected by $F'_{Sh-Mod}$, due to the better *precision* values. Moreover, since the *F*1-score is equal to the harmonic mean between *TPR* and *precision*, $F^*_{Sh-Mod}$ also gets better results for this metric.

For the highest values of $\mu$, it is more difficult to distinguish the non-bridge from the bridge nodes, which increases the number of *false positives*. So, statistics *FPR*, *AUC*, *precision* and *F1* decreases. As in the previous experiments, both models $F^*_{Sh-Mod}$-LSE and $F'_{Sh-Mod}$-LSE obtain

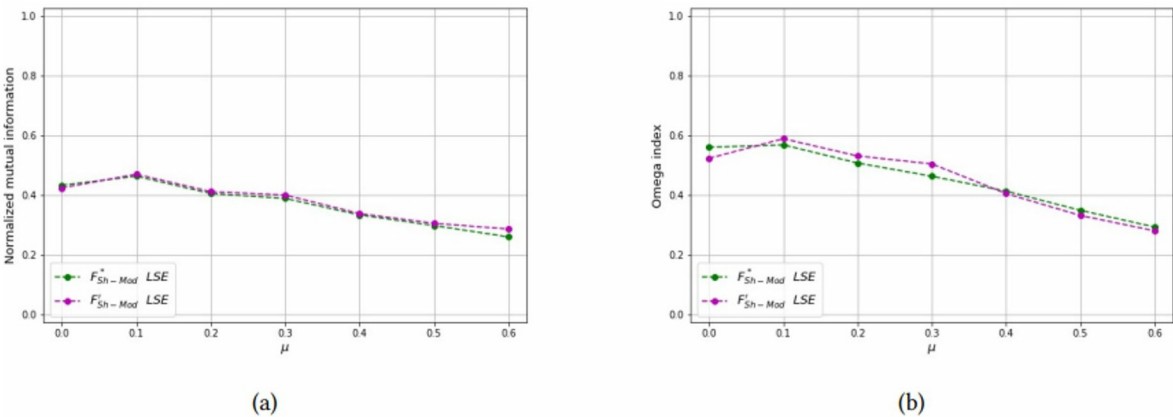

**Fig 21. Test results on overlapping communities, parameters $N$ = 500, $n_c$ = 25, $p$ = 2, $\mu_o$ = 0.6, $N_o$ = 50.** (a) Average *NMI* for each solution method, (b) Average *Omega* for each solution method.

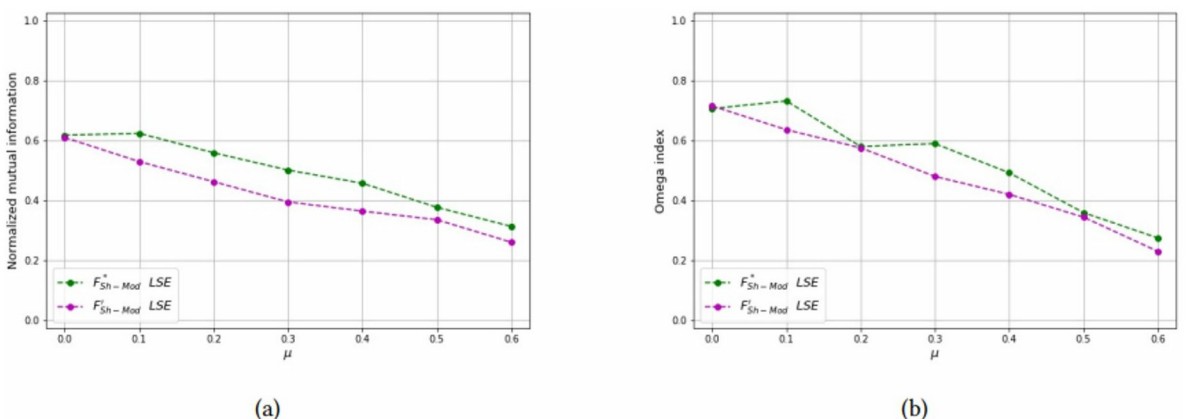

**Fig 22. Test results on overlapping communities, parameters $N$ = 500, $n_c$ = 25, $p$ = 3, $\mu_o$ = 0.7, $N_o$ = 20.** (a) Average *NMI* for each solution method, (b) Average *Omega* for each solution method.

the best results when $\mu$ is near 0 and when a bridge node belongs to many communities, as it is easier to be detected. In conclusion, $F^*_{Sh-Mod}$ detects more bridge nodes, so it obtains the highest *TPR*, but at the cost of incurring in a higher number of *false positive* too, which leads to the worst *accuracy*.

## Conclusion

In this paper, we proposed an Integer Linear Programming model to detect overlapping communities in a network. Our contribution identifies communities as stable coalitions and then we select the best of them with an optimization model. Peculiar to this approach is the definition of a weighted graph connection game and its characteristic function. Moreover, we introduced a null hypothesis in the spirit of the modularity function, [1]: We have compared the community node similarity of the actual graph with the node similarity of a random graph with no embedded communities, and in this way we could define a new similarity measure. Then, these similarities are used to define the non-convex cooperative game and the objective

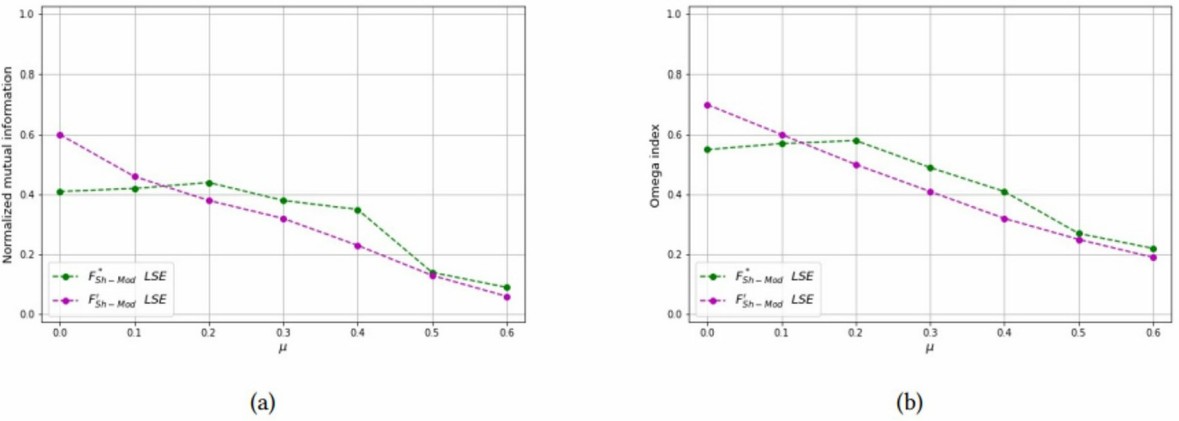

**Fig 23. Test results on overlapping communities, parameters $N = 1000$, $n_c = 50$, $p = 2$, $\mu_o = 0.6$, $N_o = 50$.** (a) Average *NMI* for each solution method, (b) Average *Omega* for each solution method.

function of a maximization problem. Nodes similarities are obtained through the application of Theorem 1, or by a simplified formula, see (21), useful to reduce the computational complexity. Computational tests show that they find similar communities.

Future research can be devoted to define stability with cooperative games others than graph connection games, and they could depend on the actual social or economic activity that is taking place on the network. We could imagine matching or voting game, to define a few, that could promptly be defined and applied to peculiar networks. Moreover, the implementation of the LSE heuristic, Algorithm 1, has been necessary to find solutions in a reasonable computation time and we found that the stability property increased the problem complexity. As stable community structures are poorly analyzed in literature, we expect that there is large room to improve our basic heuristic subroutines.

Finally, our extension of the procedure proposed in [16] to generate controlled overlapping communities can be used to validate any other method or algorithm. Testing algorithms is a big challenge and the generation of heterogeneous networks makes the comparison between algorithms easier. However, the wide combinations of parameters complicates the issue, advancing the need for a general methodology to select the most appropriate scenarios.

## Supporting information

**S1 Appendix. Appendix: Random networks generation.** Random network generator based on [16] benchmark.
(PDF)

## Author Contributions

**Conceptualization:** Stefano Benati, Justo Puerto, Antonio M. Rodríguez-Chía, Francisco Temprano.

**Formal analysis:** Stefano Benati, Justo Puerto, Antonio M. Rodríguez-Chía, Francisco Temprano.

**Investigation:** Stefano Benati, Justo Puerto, Antonio M. Rodríguez-Chía, Francisco Temprano.

**Methodology:** Stefano Benati, Justo Puerto, Antonio M. Rodríguez-Chía, Francisco Temprano.

**Project administration:** Stefano Benati, Justo Puerto, Antonio M. Rodríguez-Chía, Francisco Temprano.

**Resources:** Stefano Benati, Justo Puerto, Antonio M. Rodríguez-Chía, Francisco Temprano.

**Software:** Stefano Benati, Justo Puerto, Antonio M. Rodríguez-Chía, Francisco Temprano.

**Supervision:** Stefano Benati, Justo Puerto, Antonio M. Rodríguez-Chía.

**Validation:** Stefano Benati, Justo Puerto, Antonio M. Rodríguez-Chía, Francisco Temprano.

**Visualization:** Stefano Benati, Justo Puerto, Antonio M. Rodríguez-Chía, Francisco Temprano.

**Writing – original draft:** Stefano Benati, Justo Puerto, Antonio M. Rodríguez-Chía, Francisco Temprano.

**Writing – review & editing:** Stefano Benati, Justo Puerto, Antonio M. Rodríguez-Chía, Francisco Temprano.

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
