## [Decision Letter · Decision Letter 0]

27 Dec 2022

PONE-D-22-15051Overlapping communities detection through weighted graph community gamesPLOS ONE

Dear Dr. TEMPRANO GARCIA,

Thank you for submitting your manuscript to PLOS ONE. After careful consideration, we feel that it has merit but does not fully meet PLOS ONE’s publication criteria as it currently stands. Therefore, we invite you to submit a revised version of the manuscript that addresses the points raised during the review process.

Based on the review carried out, my recommendation is minor revision.

We look forward to receiving your revised manuscript.

Kind regards,

José F. Vicent, Ph.D.

Academic Editor

PLOS ONE

https://journals.plos.org/plosone/s/fileid=ba62/PLOSOne_formatting_sample_title_authors_affiliations.pdf.2.

“This research has been partially supported by the Agencia Estatal de Investigacion (AEI) and ´ 651 the European Regional Development Fund (ERDF): PID2020-114594GB–{C21,C22}; 652 P18-FR-1422, FEDER-UCA18-106895; and Fundacion BBVA: project NetmeetData (Ayudas ´ 653 Fundacion BBVA a equipos de investigaci ´ on cient ´ ´ıfica 2019).”

“The author(s) received no specific funding for this work”

Additional Editor Comments:

Based on the review carried out, my recommendation is minor revision.

The authors proposal is a new model (an integer linear programming model) to detect the overlapping communities in a network using cooperative games and mathematical programming. The main contribution is clear, and the mathematical notation is correct.

The experimental setup is good, but the chosen datasets are very small (less 100 nodes). I suggest to add larger datasets to test the scalability of the proposal.

Reviewers' comments:

Reviewer's Responses to Questions

**Comments to the Author**

1. Is the manuscript technically sound, and do the data support the conclusions?

Reviewer #1: Yes

2. Has the statistical analysis been performed appropriately and rigorously? 

Reviewer #1: Yes

3. Have the authors made all data underlying the findings in their manuscript fully available?

Reviewer #1: No

4. Is the manuscript presented in an intelligible fashion and written in standard English?

Reviewer #1: Yes

5. Review Comments to the Author

Reviewer #1: The authors proposal is a new model (an integer linear programming model) to detect the overlapping communities in a network using cooperative games and mathematical programming. The main contribution is clear, and the mathematical notation is correct.

The experimental setup is good, but the chosen datasets are very small (less 100 nodes). I suggest to add larger datasets to test the scalability of the proposal.

6. PLOS authors have the option to publish the peer review history of their article (what does this mean?). If published, this will include your full peer review and any attached files.

Reviewer #1: No

---

## [Author Response · Author response to Decision Letter 0]

7 Mar 2023

Dear reviewers, 

following the journal requirements, we have addressed the additional requirements:

(1) We have adapted our manuscript to the PLOS ONE’s style.

(2) All codes and datasets needed to reproduce our experimentation are

now shared.

(3) Funding Information and Financial Disclosure sections have been 

corrected and matched.

(4) Acknowledgements Section has been removed.

Following referees’ suggestions:

(1) The setup is good, but the chosen datasets are very small (less 100 

nodes). I suggest to add larger datasets to test the scalability of the 

proposal.

We have added three real datasets with 115, 198 and 453 nodes that are 

commonly used in literature. The ILP model is not able to certify optimality 

for instances with these sizes, so they have been solved by our LSE 

heuristic. Moreover, in Results and discussion section we solved random 

instances of 500 and 1000 nodes using the same algorithm.

---

## [Decision Letter · Decision Letter 1]

20 Mar 2023

Overlapping communities detection through weighted graph community games

PONE-D-22-15051R1

Dear Dr. TEMPRANO GARCIA,

We’re pleased to inform you that your manuscript has been judged scientifically suitable for publication and will be formally accepted for publication once it meets all outstanding technical requirements.

Kind regards,

José F. Vicent, Ph.D.

Academic Editor

PLOS ONE

Additional Editor Comments (optional):

After the changes made and with the review done, I think the paper should be accepted.

Reviewers' comments:

Reviewer's Responses to Questions

**Comments to the Author**

1. If the authors have adequately addressed your comments raised in a previous round of review and you feel that this manuscript is now acceptable for publication, you may indicate that here to bypass the “Comments to the Author” section, enter your conflict of interest statement in the “Confidential to Editor” section, and submit your "Accept" recommendation.

Reviewer #1: All comments have been addressed

2. Is the manuscript technically sound, and do the data support the conclusions?

Reviewer #1: Yes

3. Has the statistical analysis been performed appropriately and rigorously? 

Reviewer #1: Yes

4. Have the authors made all data underlying the findings in their manuscript fully available?

Reviewer #1: Yes

5. Is the manuscript presented in an intelligible fashion and written in standard English?

Reviewer #1: Yes

6. Review Comments to the Author

Reviewer #1: All the suggestions have been corrected by the authors. I recommend its publication at the current form.

7. PLOS authors have the option to publish the peer review history of their article (what does this mean?). If published, this will include your full peer review and any attached files.

Reviewer #1: No

---

## [Editor Report · Acceptance letter]

24 Mar 2023

PONE-D-22-15051R1 

Overlapping communities detection through weighted graph community games  

Dear Dr. Temprano:

I'm pleased to inform you that your manuscript has been deemed suitable for publication in PLOS ONE. Congratulations! Your manuscript is now with our production department. 

Kind regards, 

on behalf of

Dr. José F. Vicent 

Academic Editor

PLOS ONE